Resource

# Identification and characterization of distinct brown adipocyte subtypes in C57BL/6J mice

Ruth Karlina[1,2,*], Dominik Lutter[2,3,*], Viktorian Miok[2,3,4], David Fischer[5], Irem Altun[1,2], Theresa Schöttl[1,2], Kenji Schorpp[6], Andreas Israel[1,2], Cheryl Cero[7], James W Johnson[7], Ingrid Kapser-Fischer[1,2], Anika Böttcher[2,8], Susanne Keipert[9], Annette Feuchtinger[10], Elisabeth Graf[11], Tim Strom[11], Axel Walch[10], Heiko Lickert[2,8], Thomas Walzthoeni[5], Matthias Heinig[5], Fabian J Theis[5,12], Cristina García-Cáceres[2,4], Aaron M Cypess[7], Siegfried Ussar[1,2,13]

**Brown adipose tissue (BAT) plays an important role in the regulation of body weight and glucose homeostasis. Although increasing evidence supports white adipose tissue heterogeneity, little is known about heterogeneity within murine BAT. Recently, UCP1 high and low expressing brown adipocytes were identified, but a developmental origin of these subtypes has not been studied. To obtain more insights into brown preadipocyte heterogeneity, we use single-cell RNA sequencing of the BAT stromal vascular fraction of C57/BL6 mice and characterize brown preadipocyte and adipocyte clonal cell lines. Statistical analysis of gene expression profiles from brown preadipocyte and adipocyte clones identify markers distinguishing brown adipocyte subtypes. We confirm the presence of distinct brown adipocyte populations in vivo using the markers EIF5, TCF25, and BIN1. We also demonstrate that loss of *Bin1* enhances UCP1 expression and mitochondrial respiration, suggesting that BIN1 marks dormant brown adipocytes. The existence of multiple brown adipocyte subtypes suggests distinct functional properties of BAT depending on its cellular composition, with potentially distinct functions in thermogenesis and the regulation of whole body energy homeostasis.**

## Introduction

Brown adipose tissue (BAT) is a comparably small adipose tissue depot. Yet, in light of pandemics of obesity and the metabolic syndrome, much attention is focused on the ability of BAT to dissipate energy in form of heat through uncoupling protein-1 (UCP1)–mediated mitochondrial uncoupling (1, 2). Increasing BAT activity results in weight loss promoting metabolic health (3, 4), whereas reduction in BAT mass or function associates with increased adiposity and the metabolic syndrome (5, 6). In mice, classical BAT depots are found predominantly in the interscapular region and around the neck (7). Murine BAT development starts around embryonic days 9.5–11.5 (8), through differentiation of a *Myf5*-positive skeletal muscle/brown adipocyte precursor population (9). In contrast, most murine white adipose tissues (WAT) appear largely after birth. *Myf5*-positive precursor cells were long considered to only differentiate into murine brown adipocytes. Recently, however, expression of *Myf5* was also found in white adipocyte precursor cells of different WAT depots (10, 11, 12), questioning the selectivity of *Myf5* positive precursors for the brown adipocyte lineage. Sebo and colleagues, reported that in addition to *Pax3*+/*Myf5*+ cells, a minor fraction of interscapular brown adipocytes derive from the central dermomyotome expressing *Pax7* and *Myf5* (13). Thus, these data suggest that more than one brown adipocyte precursor cell population might exist, albeit the current dogma suggests a single precursor population (12).

In addition to classical BAT, chronic cold exposure, *β*-adrenergic agonist treatment, as well as other factors, such as nutrients, trigger the formation of so-called beige, brown-like, adipocytes in murine WAT (14, 15). These cells are similar to brown adipocytes, expressing UCP1 and dissipating energy through mitochondrial uncoupling, but differ in gene expression (16, 17) and specific functions (18, 19) from classical brown adipocytes.

[1]Research Group Adipocytes and Metabolism, Institute for Diabetes and Obesity, Helmholtz Zentrum München, Neuherberg, Germany   [2]German Center for Diabetes Research (DZD), Neuherberg, Germany   [3]Computational Discovery Research Unit, Institute for Diabetes and Obesity, Helmholtz Zentrum München, Neuherberg, Germany   [4]Institute for Diabetes and Obesity, Helmholtz Diabetes Center, Helmholtz Center Munich, Neuherberg, Germany   [5]Institute for Computational Biology, Helmholtz Center Munich, Neuherberg, Germany   [6]Assay Development and Screening Platform, Institute for Molecular Toxicology and Pharmacology, Helmholtz Zentrum München, Neuherberg, Germany   [7]Diabetes, Endocrinology and Obesity Branch, National Institutes of Health, Bethesda, MD, USA   [8]Institute for Diabetes and Regeneration Research, Helmholtz Center Munich, Neuherberg, Germany   [9]Department of Molecular Biosciences, Stockholm University, Stockholm, Sweden   [10]Research Unit Analytical Pathology, Helmholtz Zentrum München, Neuherberg, Germany   [11]Institute for Human Genetics, Helmholtz Zentrum München, Neuherberg, Germany   [12]Department of Mathematics and School of Life Sciences Weihenstephan, Technical University of Munich, Munich, Germany   [13]Department of Medicine, Technische Universität München, Munich, Germany

Correspondence: siegfried.ussar@helmholtz-muenchen.de; dominik.lutter@helmholtz-muenchen.de
*Ruth Karlina and Dominik Lutter contributed equally to this work

Detailed understanding of the developmental origins of murine brown and beige adipocytes is essential to foster the translation of basic murine research data to human physiology, as the nature of adult human brown adipocytes remains a matter of active debate. Comparative gene expression analysis initially suggested that human brown adipocytes resemble a gene expression signature more closely related to murine beige adipocytes (17, 20, 21). However, additional studies did not observe this correlation and suggested higher similarities between murine and human brown adipocytes (16). Moreover, several human BAT depots exist that show different histological and functional features and potentially distinct developmental origins (22, 23, 24). Indeed, Xue and colleagues described heterogeneity in UCP1 expression in immortalized single cell clones derived from human BAT (25). However, it remains to be determined if this heterogeneity depends on genetic differences between donors, environmental factors or the existence of distinct human brown adipocyte lineages. The latter is appealing, as the existence of multiple human brown adipocyte lineages, and potential differences in the inter-individual contribution of these lineages to human BAT, could explain some of the discrepancies in the comparison of human BAT to murine beige and brown adipocytes. These findings also raise the question if murine BAT is, as commonly believed, composed of one brown adipocyte lineage, or a mixture of brown adipocytes derived from multiple lineages. Indeed, housing mice at thermoneutrality, mimicking a physiological state more closely related to humans, results in murine BAT resembling many of the histological and functional features of human BAT (26). Importantly, murine BAT at thermoneutrality also presents histologically as a mixture of unilocular and multilocular adipocytes, resembling white and brown/beige adipocytes, respectively. The heterogeneous response of murine BAT in response to changes in ambient temperature further supports functional or developmental heterogeneity among brown adipocytes within one depot. This was recently supported by Song and colleagues, who identified, using genetic lineage tracing, single-cell RNA sequencing (scRNAseq) and other methods, the presence of UCP1 low and high murine brown adipocyte subtypes within interscapular BAT (27).

Similar to this, we observe heterogeneity in UCP1 expression of murine interscapular BAT in vivo. Using scRNAseq, we characterize the composition of the stromal vascular fraction (SVF) of murine BAT. Computational analysis of the scRNAseq data, however, revealed that identified brown preadipocyte clusters represent predominantly different stages of adipocyte differentiation rather than distinct preadipocyte subtypes. Using an alternative approach to identify distinct brown adipocyte subtypes in mice, we performed a detailed phenotypic and computational analysis of brown preadipocyte and adipocyte clones derived from the interscapular murine BAT. Using nonlinear manifold learning approaches we identify a set of potential brown adipocyte subtype markers with distinct correlations to various known markers of BAT. We identify EIF5, TCF25, and BIN1 as markers for brown adipocyte subsets, as well as their precursors, and correlate their expression with UCP1 in brown adipocytes in vivo.

Functionally we demonstrate that loss of *Bin1* increases UCP1 expression and mitochondrial respiration. Thus, our data strongly support the existence of multiple murine brown preadipocyte subtypes differentiating into potentially functionally distinct brown adipocytes and identify BIN1 as a negative regulator of thermogenesis in brown adipocytes.

# Results

## scRNAseq reveals composition and heterogeneity within the SVF of murine BAT

UCP1 is the most specific and selective marker for murine and human BAT (28). Comparison between interscapular and subscapular BAT in adult male C57BL/6J mice, did not show differences in *Ucp1* expression between these depots (Fig S1A). However, as recently reported (27), immunofluorescence stainings of adult murine interscapular BAT revealed a mosaic of UCP1 expression throughout the tissue (Fig 1A). Quantification of UCP1 immunoreactivity and grouping into quartiles of UCP1 content showed an even split between high and low UCP1 expressing brown adipocytes (Figs 1B and S1B). To obtain insights into composition and heterogeneity of the SVF of BAT, including brown preadipocytes, we performed scRNAseq from the SVF of interscapular BAT of 8-wk-old male C57BL/6J mice.

Louvain clustering of the gene expression of 4,013 cells identified 12 distinct clusters (Fig 1C and Supplemental Data 1). Using sets of marker genes we identified several immune cell populations (Fig 1C and Supplemental Data 1). Moreover, using *Pdgfra*, *Fgf10*, *Zfp423*, and *Cd34* expression, as well established markers for preadipocytes, we identified two distinct brown preadipocyte clusters (Figs 1C and S1C and Supplemental Data 1). We also identified a putative mature brown adipocyte cluster (cluster 7) with high *Fabp4* and *Pparg* expression (Figs 1C and S1C), which is further supported by the high count of mitochondrial genes in this cluster (Supplemental Data 1). Analysis of the preadipocytes only (clusters 0 and 8) identified four clusters (Fig 1D). Differential gene expression and Kyoto Encyclopedia of Genes and Genomes (KEGG)-enrichment analysis identified a number of differentially expressed genes and related pathways (Fig 1E and Table S1). Clusters 0, 1 and 2 were assigned with a number of distinct pathways, whereas cluster 3 shared most of its enriched pathways except "Wnt signaling" with other clusters. Cluster 0 associated with peroxisome proliferator-activated receptor (PPAR) and Apelin signaling, both important for terminal brown adipocyte differentiation (29, 30), whereas clusters 1–3 showed an enrichment in pathways, such as WNT, TNF, relaxin, and TGFβ signaling that are generally associated with the inhibition of adipogenesis (31, 32, 33). Interestingly, cluster 3 shared most pathways with other clusters, regardless of their distance in the louvain plots. Pseudotime mapping of the preadipocytes largely replicated the louvain clustering, albeit it revealed two distinct developmental branches, with one common precursor (Fig 1F). Analysis of the expression of known markers during adipogenesis identified cluster 2 (enriched in *Cd34* expressing cells) as the most undifferentiated state, with transition states (expressing *Cebpd* and *Cebpb*) representing clusters 1 and 3. Cluster 0, as suggested by the KEGG-enrichment analysis, showed enrichment in cell expressing terminal differentiation markers such as *Cebpa*, *Pparg*, *Fabp4*, and

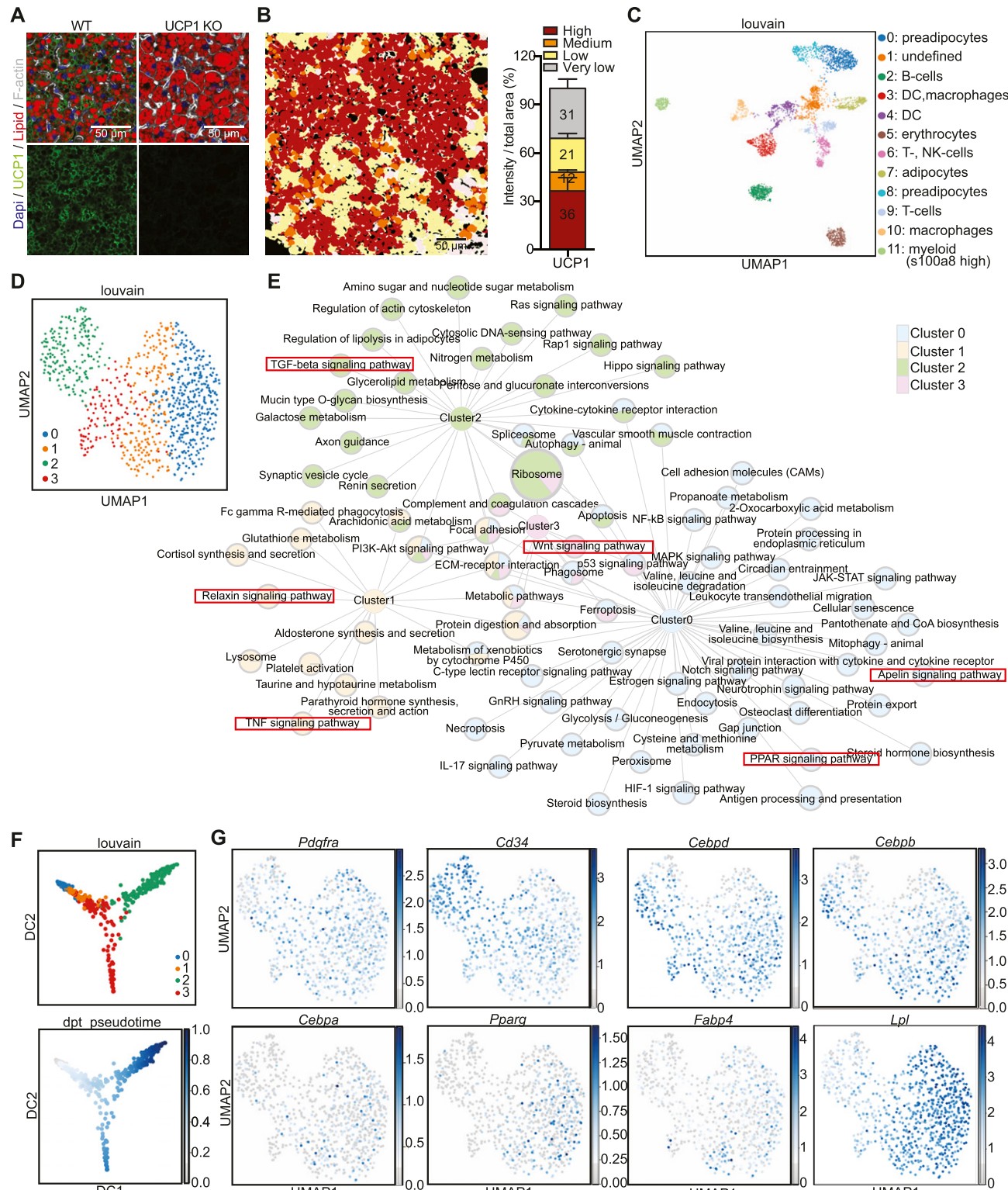

**Figure 1. scRNAseq identifies distinct stages of brown adipocyte differentiation.**
**(A)** Immunofluorescence staining of UCP1 (green), F-actin (white), lipid droplets (red), and DAPI (blue) from WT and UCP1 knockout mice. **(B)** Quantifications of UCP1 content in individual brown adipocytes in brown adipose tissue normalized to total area (right panel, n = 11). Lipid droplets are included into area measured, and F-actin was used to distinguish each cell. **(C)** UMAP computed on full processed single-cell RNA-seq data set with annotated louvain clusters superimposed. **(D)** UMAP computed on set of preadipocytes with louvain subclustering. **(E)** Network of KEGG-enriched pathways of preadipocyte clusters. Significantly enriched pathways are connected to the respective cluster nodes. Size of the pathway nodes and proportion of the node-pie-chart refer to $-\log_{10}$ of enrichment *P*-value. **(D)** Node colors refer to single-cell cluster identified in (D). **(F)** Diffusion map dimensionality reduction of preadipocytes colored by Louvain clusters (top panel) and pseudotime (lower panel).

*Lpl* (Fig 1G). Thus, these data strongly suggest that the four clusters identified by scRNAseq do not represent individual preadipocyte subtypes, but rather distinct differentiation stages of preadipocytes. This further suggests that the differences in gene expression upon differentiation mask differences in gene expression between potential brown preadipocyte lineages.

### Murine brown adipocyte clones differ in brown, beige and white marker gene expression

To overcome the limitations of a static experimental setup such as scRNAseq, we established 67 clonal cell lines, derived from the SV40 large-T immortalized SVF of three 8-wk-old male C57BL/6J mice. Most cell lines, when differentiated for 8 d, accumulated lipids, albeit to different extents, as quantified by Oil Red O (ORO; Fig 2A) and LipidTOX stainings (Fig 2B and C). Morphological analysis of the preadipocyte and adipocyte clones using LipidTOX and phalloidin (to stain the F-actin cytoskeleton) did not reveal major differences in shape or lipid droplet size between the clones, with the exception of clones 1A6 and 1D5, which had smaller lipid droplets when correlated with the overall lipid content (Fig 2C). Correlation between lipid content (ORO) and *Pparg* expression at day 8 of differentiation (Fig 2D) showed a strong correlation between lipid accumulation and *Pparg* expression. However, there were several clones (1B1, 1B3 and 1B6, 2C12, 3C7 and 3D6) that diverged and showed higher *Pparg* expression than anticipated from the lipid content. In line with the function of PPARG in driving brown adipocyte selective gene expression, clones 1B3, 1B6, 2B11, 2C7, 2C12, 3C7, and 3D6 showed highest *Ucp1* expression compared to all other clones (Fig 3A).

Next, we tested expression of additional markers for brown (PRDM16 and PPARGC1A), beige (TMEM26 and CD137) adipocytes (17, 34) and performed unsupervised hierarchical clustering (Euklidean distance and Ward's linkage method) of the data to see if the clonal cell lines could be grouped using these markers. For comparison, we also included in vitro differentiated brown, beige and subcutaneous white adipocytes in the analysis (Figs 3B and S2A). 19 cell lines clustered with brown adipocytes, 18 with beige, and 15 with white adipocytes, with two additional clusters containing six and nine cell lines with either very high or very low thermogenic gene expression, respectively. The latter cluster was largely defined as *Cd137*high and low for all other markers (Fig 3B). The 67 different cell lines expressed very different combinations of brown, beige and white adipocyte markers. Expression of *Prdm16* and *Ppargc1a*, two key factors determining differentiation of preadipocytes towards brown adipocytes (9, 35, 36) were largely co-expressed with *Ucp1* (Fig 3B). This was also observed in a correlation between *Prdm16*, *Ppargc1a*, and *Ucp1*, respectively, while the correlation between *Pparg* and *Ucp1* expression was weaker (Fig 3C). However, there were several clones that revealed a surprising dissociation of expression between these markers for brown adipocytes. The beige adipocyte markers *Cd137* and *Tmem26* were not expressed in the

same pattern, with highest expression of these genes in different non-overlapping cell lines, which is in line with a recent report describing CD137 as a negative regulator of beige adipocyte function (37). Moreover, RT-qPCR analysis for *Adiponectin*, *Cd36*, *Fabp4*, *Glut4*, and *Hsl* from 64 out of the 67 cell clones (not including three cell lines with lowest differentiation capacity: 2D5, 3A9, and 3A10) revealed heterogeneity in general adipocyte marker gene expression (Fig S2B). Interestingly, inter-individual differences were smaller than differences between clonal lines of the same mouse. This suggests that there is developmental heterogeneity between preadipocytes.

Based on the observed differences in brown, beige and white characteristics of the clonal cell lines, we tested cellular response to acute (0.5 μM for 3 h) β3-adrenergic receptor agonist (CL-316,243) treatment, with respect to *Ucp1* gene expression (Fig 3D). *Ucp1* expression was induced in most cell lines by acute CL-316,243 treatment, except 1A3, 1C6, 2B4, 2B5, 2B10, 2C6, 2D5, 3A10, 3C4, and 3D9, which were also the least differentiated cell lines (Figs 3D and 2A). Thus, the data obtained from these 67 clonal brown adipocyte cell lines strongly suggested functional and developmental heterogeneity.

### RNAseq expression profiling reveals differences in brown preadipocytes and adipocytes

To obtain a more detailed view on the transcriptional differences of the cell lines, we performed RNA sequencing (RNAseq) from all 20 undifferentiated and differentiated cell lines of mouse 1, that all showed comparable proliferation rates (Fig S3), indicating no differences in the effects of SV40 immortalization. After pre-processing and filtering (38) a total of 9,483 for undifferentiated and 10,363 genes for differentiated brown adipocytes remained for further analysis.

Unsupervised hierarchical clustering for pre- and differentiated adipocytes did not reveal any common pattern of conserved cell identities between undifferentiated and differentiated cells (Fig 4A). This was also confirmed by PCA, which did not indicate any conclusive clustering or pattern that would allow for the identification of distinct brown adipocyte lineages (Fig 4B). To compare our data set with the gene expression profiles of whole WATs and BATs and different cell lines, we analyzed our data sets using the ProFAT database (39). This online tool provides a relative estimation of the BAT characteristics of the analyzed samples by comparing gene expression data to a reference database of white and brown murine and human adipose tissue samples (39). Using k-means clustering, we identified three cell clusters with specific features compared with BAT in both the undifferentiated and differentiated states. One cluster did not show characteristics of BAT in either the undifferentiated or differentiated states (circles); the second acquired brown fat characteristics upon differentiation (triangles), whereas the third one showed BAT characteristics in both undifferentiated and differentiated cells (squares) (Fig 4C). Because all cell lines were derived from BAT, we display the data by similarity to brown fat "BATness," scaled from 0 (low) to 1 (high). We performed pairwise

**(D, G)** Expression of *Pdgfra*, *Cd34*, *Cebpd*, *Cebpb*, *Cebpa*, *Pparg*, *Fabp4*, and *Lpl* in preadipocytes shown in (D). The expression values are size factor normalized and log-transformed.
Source data are available for this figure.

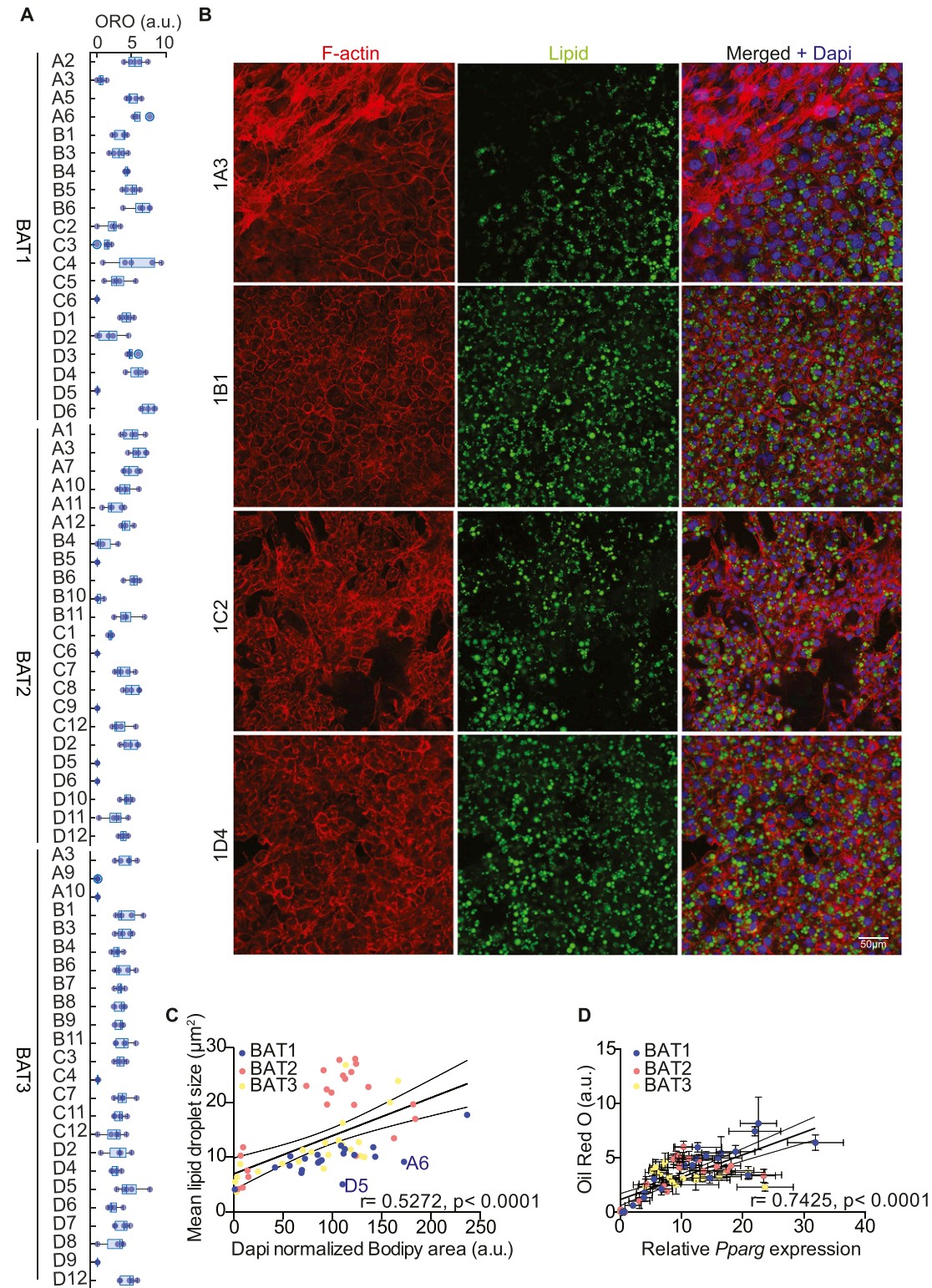

**Figure 2. Heterogeneity in differentiation capacity and lipid accumulation of brown adipocyte clones.**
**(A)** Quantification of relative lipid accumulation measured by Oil Red O staining at day 8 of differentiation from 67 immortalized brown preadipocyte clones (n = 4–6; mean OD normalized to DAPI ± SEM). **(B)** Representative images of differentiated cell lines, stained with F-actin (red), lipid droplets (green), and DAPI (blue). **(C)** Correlation between mean lipid droplet size and lipid area normalized by the number of nuclei per clone (n > 200), calculated from pictures shown in Supplemental Data 2. Values are mean of different area scanned per clones (n = 9). **(D)** Correlation analysis of *Pparg* expression and lipid accumulation. The values were mean ± SEM (n = 5). Source data are available for this figure.

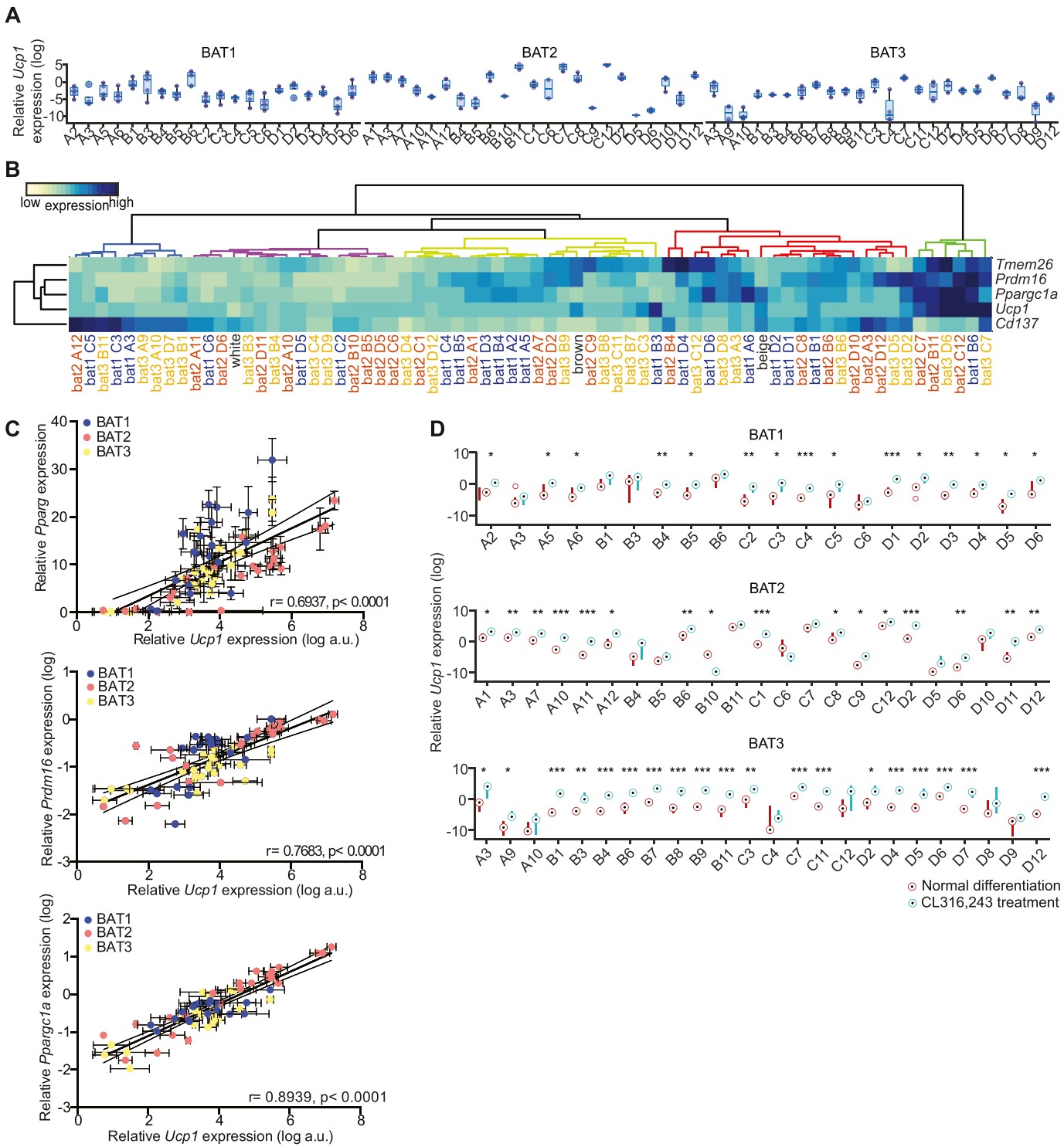

**Figure 3. Brown adipocyte clones heterogeneously express brown, beige, and white fat markers.**
**(A)** mRNA expressions of *Ucp1* at day 8 of differentiation (n = 2–5). **(B)** Heat map of mRNA expression for different adipocyte markers in all 67 clones, and in vitro differentiated white, beige and brown adipocytes at day 8 of differentiation (n = 3–8). Median expression for each cell line was transformed to log scale and gene wise z-scores were computed independently for each cell line. Dendrogram colors denote identified adipocyte clone cluster. **(C)** Correlation of *Ucp1* with *Pparg* (left panel), *Prdm16* (middle panel), and *Ppargc1a* (right panel). The values were mean ± SEM, and log transformed for *Ucp1*, *Prdm16*, and *Ppargc1a* (n = 4–5). Gene expression was normalized to *Tbp*. **(D)** *Ucp1* expression of controls and cells treated for 3 h with 0.5 μM CL-316,243 (n = 2–5).
Source data are available for this figure.

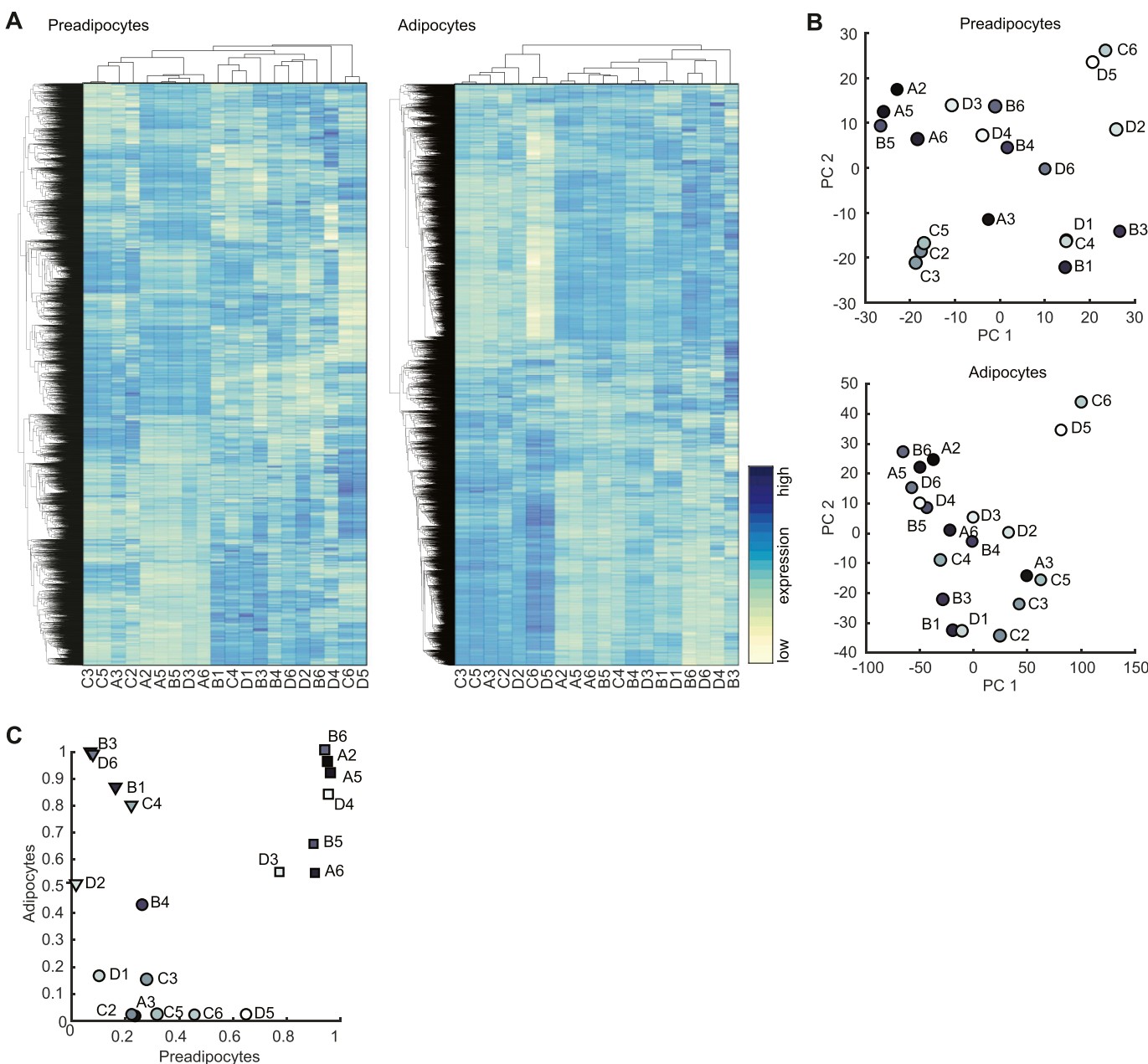

Figure 4. **Transcriptional profiling of selected BAT clones. (A)** Heat map of all 9,483 expressed genes in preadipocyte. Gene expression in rows was z-score normalized, columns refer to cell lines (left panel). Hierarchical clustering heat map of all 10,363 expressed genes in differentiated adipocytes (right panel). **(B)** PCA plot of the first two PC of preadipocytes expression data (upper panel) and differentiated adipocytes (lower panel). **(C)** Scatterplot comparison of the estimated brown adipose tissueness from pre- and differentiated adipocyte cell lines. Dots, triangles, and square indicate the three clusters identified by k-means clustering.

comparisons for each undifferentiated and differentiated sample of each cell line (Fig 4C).

### Laplacian eigenmap–based dimensionality reduction reveals distinct brown adipocyte gene expression signatures

This classification indicated that there are groups of cell lines with distinct and shared molecular characteristics. However, the ProFAT-associated clusters did not correlate with clusters in the hierarchical

clustering of the RNAseq data. Furthermore, based on the previous analysis, we were neither able to estimate the number of brown adipocyte lineages among our clones, nor could we identify marker genes (features) allowing us to group the cell lines into lineages.

Nevertheless, the RNA expression data and ProFAT correlations also suggested cellular heterogeneity in murine BAT. Consequently, we applied nonlinear dimensionality reduction techniques to uncover hidden patterns, aiming to further classify our cell lines. To avoid a bias based on differences in differentiation capacity, and to

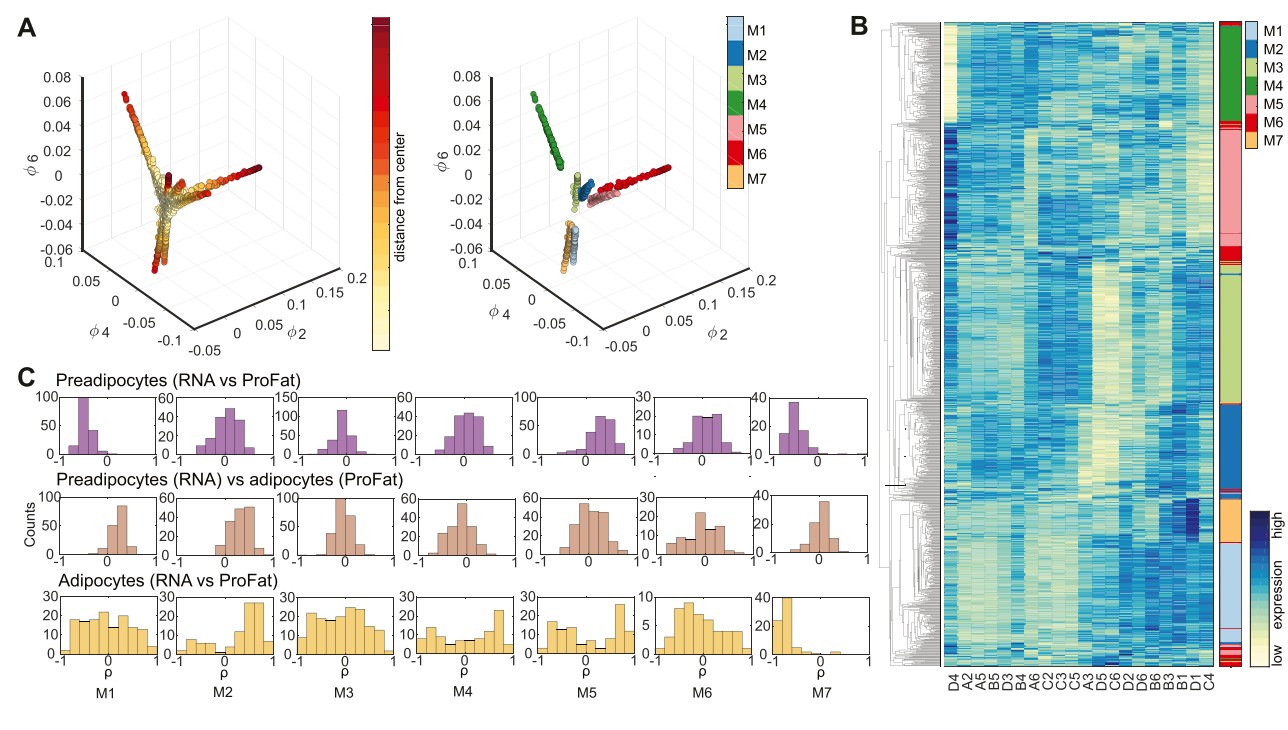

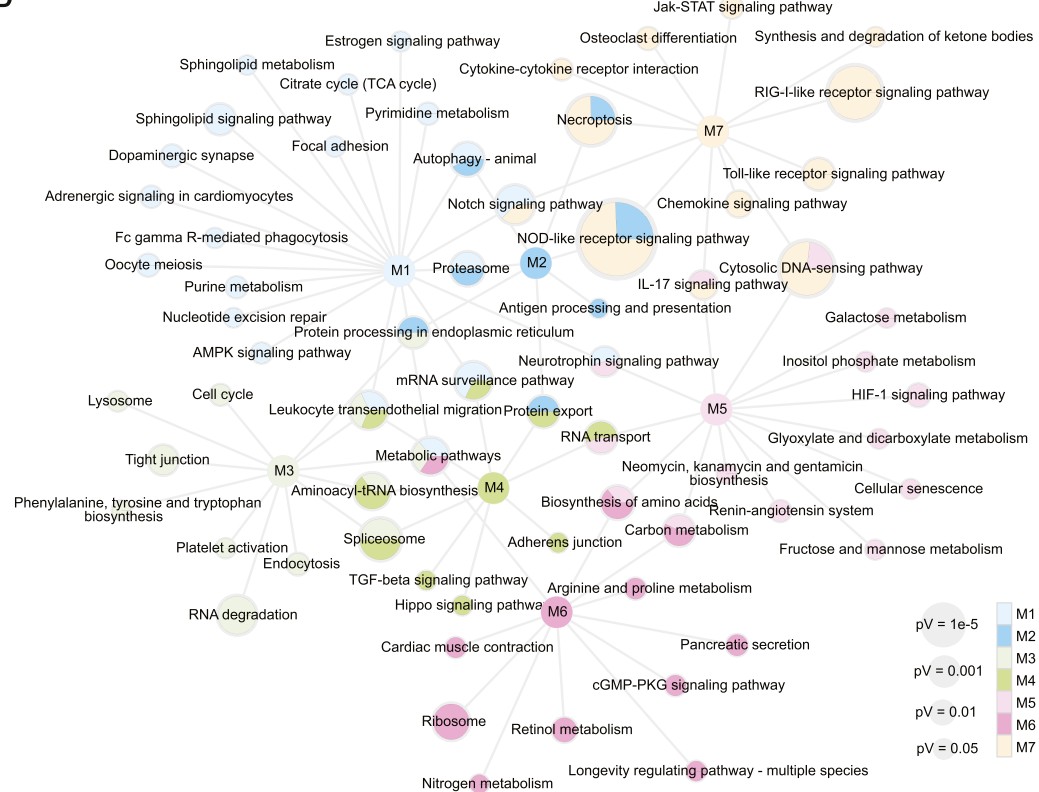

**Figure 5. Laplacian eigenmap–based extraction of expression modules.**
**(A)** Brown preadipocyte gene expression in Laplacian eigenmap–derived low dimensional space. Expression of the three eigenvectors φ2, φ4, and φ6. Distance from estimated center is color-coded (left panel). After removal of genes with low variance module membership is coded in colors (right panel). **(B)** Heat map of hierarchical clustered module genes in rows. Gene expression was z-score normalized. Module membership is indicated by the right color bar. **(C)** Distribution of correlation coefficients of the gene-wise comparison of expression to estimated brown adipose tissue (BAT)ness for each module. Upper line: preadipocyte gene expressions versus estimated BATness of preadipocytes from ProFat database. Middle line: preadipocyte expressions versus differentiated BATness from ProFat database. Lower line:

define distinct precursor populations, we restricted our analysis to the RNAseq data from preadipocytes. After applying Laplacian eigenmaps, the projected gene expression revealed seven distinct gene sets, from which we generated seven gene expression modules (GEMs) M1-M7, with sizes between 76 and 227 genes (Fig 5A and Table S2).

We next analyzed if the genes defining the seven GEMs allow defining distinct cellular populations. Almost all of the seven GEMs showed distinct gene expression patterns, only module M6 split up in small subgroups distributed all across the clustering tree (Fig 5B). Modules M1-M5 and M7 showed unique expression patterns; thus, each of them allowing for sub classification of the clonal cell lines. However, at this point, the data did not support preference of one module over the other. Furthermore, PCA of GEM gene expression of both undifferentiated and differentiated brown adipocyte clones did not reveal clear populations (Fig S4). Thus, we compared module specific expression patterns to estimated BATness from the ProFat database. The ProFat database allows comparison of gene expression data to gene expression signatures of human and murine brown adipocytes. Thus, the correlation of module specific gene expression to BATness will show if the gene expression signatures within these modules positively or negatively correlate with a brown adipocyte gene expression signature.

To this end, we correlated the individual gene expression of each preadipocyte module to the estimated BATness of either the undifferentiated (Fig 5C upper panel) or differentiated (Fig 5C middle panel) cell lines, as well as gene expression and estimated BATness of adipocytes (Fig 5C lower panel) and summarized the correlation coefficients for each gene within the individual modules. In preadipocytes (upper panel), M1 and M7 contain genes that negatively correlate with BATness (mean shifted leftwards), whereas gene expression of M5 is positively correlated to BAT (mean shift rightwards). M2–4 and M6 did not show characteristics that correlate with brown adipocytes. Conversely, genes in M1 and M7 showed a trend to positively correlate with BATness in differentiated adipocytes, similar to M2, whereas all others showed a more random distribution. KEGG-enrichment analysis of each GEM showed associations with diverse pathways and little overlap between the modules (Fig 5D and Table S3), except M2, which shared most of its enriched pathways except "antigen processing and presentation" with other modules. M1 showed highest association with sphingolipid signaling and metabolism as well as autophagy and notch signaling, whereas M7, the other WAT-associated module was characterized by pathways associated with immune cell signaling. Conversely, M5, with highest BAT correlation, associated with carbohydrate and amino acid metabolism.

Taken together, these data showed that the identified GEMs contain information to distinguish our clonal cell lines based on immune signaling, cell adhesion and migration, RNA and amino acid metabolism, and fuel processing.

## EIF5, TCF25, and BIN1 mark subpopulations of brown preadipocytes and adipocytes in vivo

To identify markers for brown adipocyte subtypes, we selected those genes with stable expression between preadipocytes and adipocytes. From the seven GEMs, we identified 57 genes showing stable expression ($P < 0.05$, Pearson correlation) between pre- and differentiated brown adipocyte clones (Fig 6A). A PCA of these genes from either undifferentiated or differentiated brown adipocyte clones did not clearly separate the cell lines into different clusters (Fig S5A). ORO quantification–based PCA on preadipocytes and adipocytes did also not show any obvious clustering of the cell lines (Fig S5B). Single gene association to browning was estimated by correlating the gene expressions to cell line specific BATness. Thus, we identified genes that potentially mark cells with distinct characteristics, indicative of distinct subtypes (Fig 6B). The two genes from M4 (*Bin1* and *Phax*) were least correlated with BATness, whereas genes from M2 were mainly correlated with BATness. The genes from the other GEMs were more evenly distributed.

We selected the eukaryotic translation initiation factor 5 (EIF5), transcription factor 25 (TCF25), and bridging integrator 1 (BIN1) to test as potential lineage markers from the stably expressed 57 genes, based on BATness, in combination with the maximum expression and fold change in our clonal cell lines as well as the availability of antibodies (Fig S5C). The eukaryotic translation initiation factor 5 (EIF5), functions to initiate protein synthesis through interaction with the 40S ribosomal subunit (40, 41). Transcription factor 25 (TCF25) has been shown to play an important role during early embryonic organogenesis (42), and its expression is decreased by age in several organs, such as kidney, heart, liver, and lung (43). Bridging integrator 1 (BIN1), or amphiphysin 2, shows the most restricted expression pattern among the three, with highest expression in skeletal muscle and brain and has been shown to regulate muscle differentiation (44).

Among those three, EIF5 was most highly associated with a classical BAT phenotype; TCF25 could not be assigned (no tendency to be more- or less-brown), whereas BIN1 was the least brown adipocyte–associated gene (Fig 6B). We observed a similar pattern when cells were correlated to lipid content (Fig S5D). The expression of *Eif5*, *Tcf25*, and *Bin1* in the differentiated cell lines was correlated with the expression, as assessed by RT-qPCR, of *Ucp1*, *Pparg*, *Ppargc1a*, and *Prdm16* (Fig 6C). *Eif5* was positively correlated with *Ucp1*, *Pparg*, *Ppargc1a*, and *Prdm16*. *Tcf25* showed no correlation with *Ucp1* and differentiation markers, whereas *Bin1* negatively correlated with *Ucp1* and the other markers (Fig 6C).

Next we tested if expression of *Eif5*, *Tcf25*, and *Bin1* showed similar associations with *Ucp1* expression in vivo. To this end, we studied gene expression in BAT of mice chronically housed at either thermoneutrality (30°C) or at 5°C (Fig 6D). As previously reported, *Ucp1* expression was increased upon chronic cold exposure when

---

differentiated adipocyte expressions versus differentiated BATness from ProFat database. **(D)** Network of KEGG-enriched pathways. Significantly enriched pathways are connected to the respective module nodes. Size of the pathway nodes and proportion of the node-pie-chart refer to –log$_{10}$ of enrichment *P*-value. Color of the nodes refers to module membership.
Source data are available for this figure.

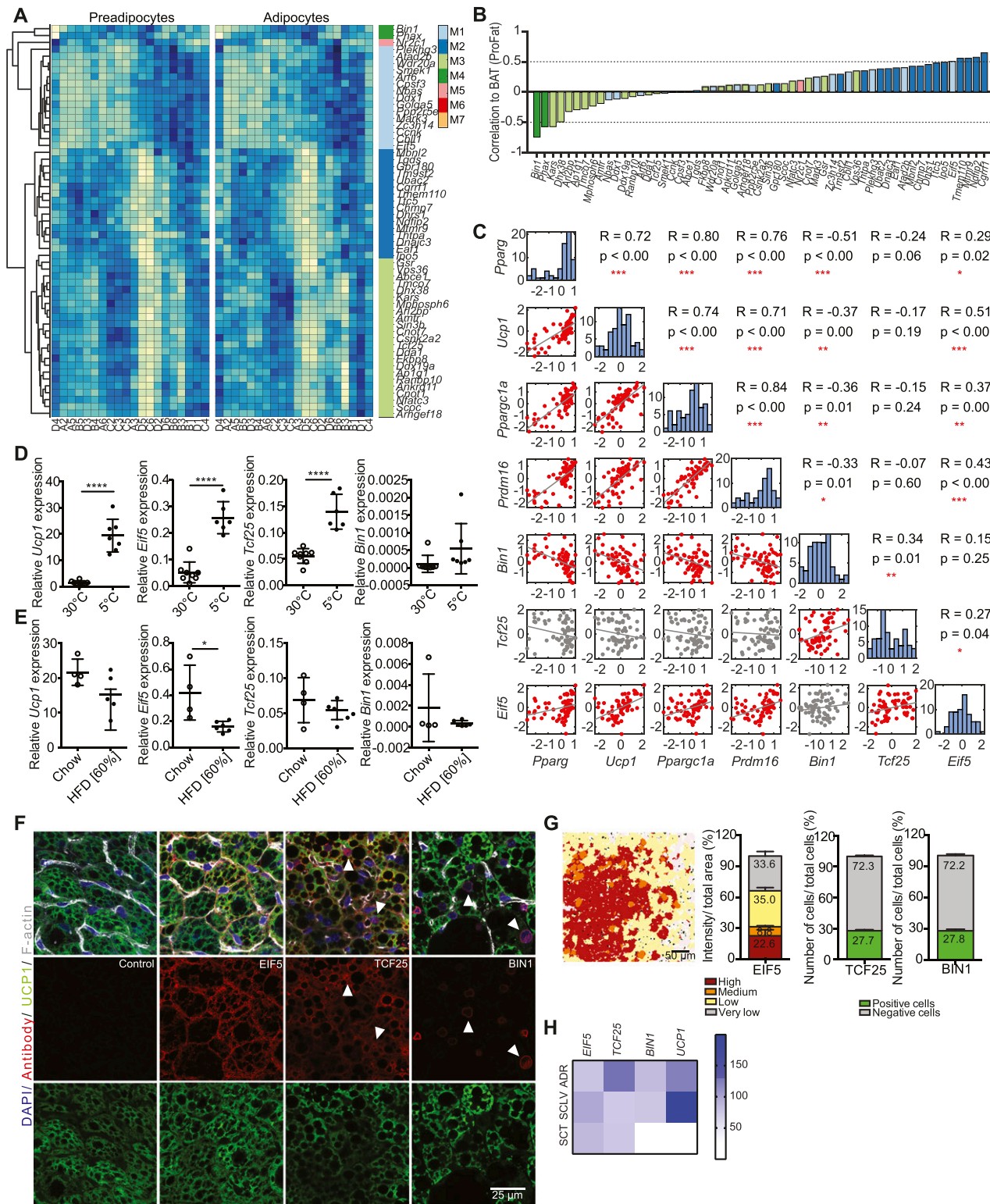

**Figure 6. EIF5, TCF25, and BIN1 mark subsets of brown adipocytes.**
**(A)** Heat maps of stably expressed genes of preadipocytes (left panel) and differentiated brown adipocytes (right panel). Module membership is indicated by the right color bar. **(B)** Correlation coefficients for stable expressed genes compared with estimated brown adipose tissue (BAT)ness. Color of bars indicates module membership. **(C)** Correlation plot for pairwise comparison of selected markers to selected stably expressed genes. Red plots denote a significant correlation. **(D)** Expression of *Ucp1*, *Eif5*, *Tcf25*, and *Bin1* in mice kept in cold or thermoneutrality (n = 6–8, data are mean expressions normalized to B2m ± SEM). **(E)** Expression analysis of *Ucp1*, *Eif5*, *Tcf25*, and *Bin1* in chow- and high-fat diet-fed mice (n = 4–6, data were mean expressions normalized to *B2m* ± SEM). **(F)** BAT co-staining for EIF5, TCF25 or BIN1 (red) and UCP1 (green),

compared with its expression at thermoneutrality. Similarly, expression of *Eif5* and *Tcf25* was increased upon chronic cold exposure, whereas Bin1 expression remained unaltered. Expression of *Ucp1* was slightly decreased in BAT of obese mice fed a high fat diet (HFD) compared with lean chow diet animals (Fig 6E). An even more pronounced reduction was observed for *Eif5*, whereas expression of *Tcf25* and *Bin1* were not altered by diet-induced obesity (Fig 6E). However, *Eif5*, *Tcf25*, and *Bin1* are also expressed in other cell types of the SVF. Thus, we cannot exclude that changes in the expression, or lack thereof, in these non (pre-) adipocyte cells impact on the whole BAT gene expression signature.

Further analysis of our scRNAseq data showed a random distribution of *Eif5*, *Tcf25*, and *Bin1* in the four preadipocyte clusters, further confirming the stable expression of these markers throughout brown adipocyte differentiation (Fig S5E).

To investigate the distribution of the selected marker genes in relation to UCP1 in mature adipocytes in vivo, we performed co-immunostainings for EIF5, TCF25, and BIN1 with UCP1 on murine BAT sections. EIF5 expression correlated with UCP1 staining intensity (Fig 6F). Following our quantification strategy applied for UCP1 (Fig 1A), we quantified EIF5 expression in individual brown adipocytes and grouped the cells according to high, medium, low, and very low expression, with around 1/4 of the brown adipocytes showing high or very high expression (Figs 6G and S5F). Nuclear or perinuclear expression of TCF25 or BIN1, respectively, was observed in ~25% of brown adipocytes (Fig 6F and G). BIN1 appeared highest expressed in low UCP1 expressing brown adipocytes, whereas TCF25 showed strongest immune reactivity in brown adipocytes with medium UCP1 expression (Fig 6F). In addition, BIN1 staining was also observed in endothelial cells. Comparative gene expression analysis in periadrenal, supraclavicular, and subcutaneous white and BAT of humans confirmed expression of *EIF5*, *TCF25*, and *BIN1* in human BAT, but also in WAT as seen in mice (Figs 6H and S5G and H).

## Loss of *Bin1* increases UCP1 expression and oxygen consumption

To functionally characterize differences of $Eif5^{high}$, $Tcf25^{high}$, and $Bin1^{high}$ brown adipocytes, we selected representative clones [B1 ($Eif5^{high}$), D1 ($Tcf25^{high}$), and D5 ($Bin1^{high}$)] using hierarchical clustering from gene expressions measured by RT-qPCR (Fig 7A). We established stable knockdown cell lines of these genes in the respective clones, which was confirmed by semi-quantitative PCR of respective gene expression on day 0 (preadipocytes) and day 8 (adipocytes) of differentiation (Fig 7B). Neither of the knockdowns impaired adipogenesis as assessed by expression of *Pparg* (Fig S6A). However, reduction of *Eif5* in the B1 clone induced *Ucp1* and *P2rx5* mRNA expression (Fig S6A), but had no significant effect on mitochondrial respiratory capacity (Fig 7C, left panel). *Tcf25* knockdown in clone D1 reduced expression of *Ucp1* and *Prdm16* and showed increased expression of *Cd137* and the white adipocyte marker *Asc1* (16) in

comparison to its shScr control (Fig S6A). Microplate-based oxygen consumption of the shTcf25 D1 clone revealed reduced maximal respiratory capacity (Fig 7C, middle panel), but no differences in mitochondrial uncoupling. Knockdown of *Bin1* in the D5 clone resulted in an up-regulation of *Ucp1* mRNA levels (Fig S6A) and showed a higher basal oxygen consumption rate (OCR). Addition of oligomycin (ATP synthase inhibitor) to the wells did not reduce the OCR of shBin1 D5, indicating full mitochondrial uncoupling (Fig 7C, right panel). Thus, expression of *Bin1* in preadipocytes and adipocytes correlates with low thermogenic capacity, and loss of *Bin1* results in increased *Ucp1* gene expression and full mitochondrial uncoupling, at least in clones with high *Bin1* expression.

We next tested the effects of stable depletion of these markers in a mixed brown preadipocyte population. To this end, we created brown preadipocyte knockdown cell lines for *Eif5* (shEif5), *Tcf25* (shTcf25) or *Bin1* (shBin1), with scrambled shRNA expressing cells (shScr) as the control. RT-qPCR analysis in preadipocytes and fully differentiated adipocytes confirmed a stable knocked-down of all three genes (Fig 7D). This was further confirmed by immunofluorescence stainings on preadipocytes (Fig 7E). As with the clonal cell lines we did not observe any differences in differentiation capacity upon loss of either of the marker genes with respect to lipid accumulation (Fig 7F and G) or expression of *Pparg* (Fig 7H, left panel), despite a reduction in *Pparg* expression in CL-316,243 treated shBin1 cells (Fig 7H), which was not observed in the knockdown of *Bin1* in the clone D5. This could suggest that loss of *Bin1* in *Bin1*-expressing cells could have a paracrine effect on neighboring precursors that do not express *Bin1*. *Cd137* mRNA levels were generally higher in the knockdown adipocytes compared with controls, whereas *Prdm16* expression was lower in shEif5 and shTcf25 lines compared with control cells (Fig S6B). In line with the clonal knockdown data, depletion of *Tcf25* resulted in increased expression of the white adipocyte marker *Asc1* and a down-regulation of the brown adipocyte marker *P2rx5* (Fig S6B). In contrast, shEif5 and shBin1 cells had significantly higher mRNA or protein levels of UCP1 without or with CL-316,243 treatment compared with the shScr adipocytes, respectively (Fig 7H and I). shBin1 cells also expressed highest levels of *P2rx5* (Fig S6B). Seahorse analysis revealed that oxygen consumption was highest in shBin1 cells (Fig 7J). Thus, *Bin1* marks a brown adipocyte population with low thermogenic capacity. Moreover, loss of *Bin1* in brown adipocytes increases UCP1 levels and mitochondrial activity suggesting that *Bin1* functions in suppressing classical brown adipocyte identity.

Taken together, these data demonstrate that using our combined approach we were able to identify a set of genes, including EIF5, TCF25, and BIN1, with preserved expression between pre-adipocytes and adipocytes that mark subsets of distinct brown adipocytes within BAT. Our data support a model where *Bin1* suppresses full mitochondrial uncoupling; marking a "dormant"

---

with F-actin (gray) and Dapi (blue) from wild-type C57BL/6J mice. Arrows indicate nuclear staining. **(G)** Representative color gradient picture of EIF5 staining on BAT (left panel), and percentage area of different EIF5 intensity (high, medium, low, and very low) normalized to total area (second panel, n = 8 sections). Quantification of TCF25- and BIN1-positive and negative cells in percentage of total cells (n = 9–18 sections). **(H)** Heat map of mRNA expression in human periadrenal (ADR), supraclavicular (SCLV), and subcutaneous (SCT) adipose tissue from six different donors for ADR and SCLV, and five donors for SCT. **(A, B)** Based on RNAseq data and remaining analysis based on RT-qPCR data. *$P < 0.05$, **$P < 0.01$, ***$P < 0.001$, ****$P < 0.0001$.
Source data are available for this figure.

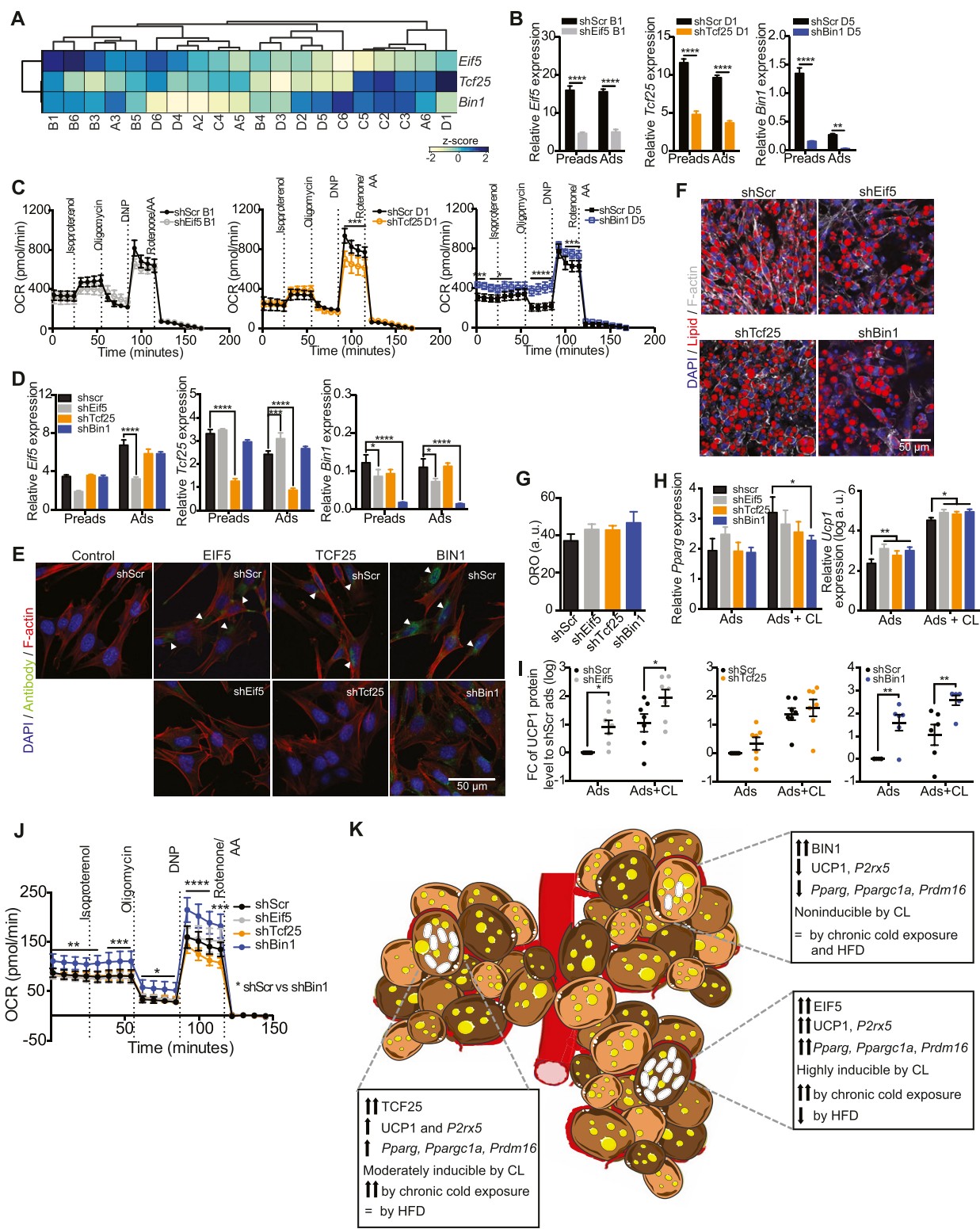

**Figure 7.  Loss of *Bin1* increases basal respiration and UCP1 protein content.**
**(A)** Heat map and hierarchical clustering of cell lines with regard to the mRNA expressions of *Eif5*, *Tcf25* and *Bin1*. **(B)** Expression of *Eif5*, *Tcf25*, and *Bin1* in control cells (shScr) of different clones (B1, D1, and D5) and the respective knockdown (shEif5 B1, shTcf25 D1, and shBin1 D5); Preads (d0 of differentiation) and Ads (d8 of differentiation) (n = 5). **(C)** Oxygen consumption rate of shScr B1 and shEif5 B1 (left panel), shScr D1, and shTcf25 D1 (middle panel) and shScr D5 and shBin1 D5 (right panel) at day 8 of differentiation measured by Seahorse (n = 4–5). **(D)** Expression of *Eif5*, *Tcf25*, and *Bin1* in control cells (shScr) and respective knock-down (shEif5, shTcf25, and shBin1); Preads (d0 of differentiation) and Ads (d8 of differentiation) (n = 5). **(E)** Immunofluorescence staining of EIF5/TCF25/BIN1 (green), F-actin (red), and DAPI (blue) on

brown adipocyte subpopulation, whereas *Eif5* and *Tcf25* positive brown adipocytes represent "classical" fully uncoupled brown adipocytes. Thus, our data strongly suggest the presence of multiple brown preadipocyte and adipocyte subtypes in mice (Fig 7K).

# Discussion

In the current study, we aimed to clarify the presence of distinct brown adipocyte lineages in mice. We find that even at room temperature, where murine BAT appears morphologically uniform, UCP1 expression within individual brown adipocytes differs greatly throughout the tissue. To this end, we tested if differences in UCP1 content are the result of developmental heterogeneity among murine brown adipocytes and not due to anatomical or environmental differences. Using scRNAseq from the SVF of adult murine BAT, we identified a diverse set of immune cell and preadipocyte clusters. Surprisingly, a more detailed analysis of the preadipocyte clusters indicated that clustering within preadipocytes was driven by differentiation rather than distinct developmental lineages. Therefore, albeit providing an interesting resource for others to study cellular composition within BAT, these scRNAseq data did not allow identification of distinct brown preadipocyte lineages. To overcome this limitation we generated, similar to a previous study on human brown adipocyte cell lines (45), 67 immortalized brown preadipocyte clones from lean adult male C57BL/6J mice housed at room temperature. Thus, as currently believed, all preadipocytes isolated from the BAT of this mouse should, in contrast to precursors isolated from human BAT (45), by definition give rise to only brown adipocytes. We show that these clones differentiate, albeit to various degrees, into lipid laden adipocytes, expressing adipogenic markers such as *Pparg*. However, these cell lines were heterogeneous with respect to *Ucp1* expression, which was not well correlated with the capacity to differentiate and accumulate lipids.

Heterogeneity between the cell lines was also apparent when comparing their response to β3-adrenergic stimulation or the expression of classical brown, beige and white adipocyte markers. Thus, similar to Shinoda et al, we observe strong differences in gene expression between the clonal cell lines (45).

To obtain more insights into the differences between the clonal cell lines, we focused on the clones from one mouse and performed RNAseq from all undifferentiated and differentiated cell lines. Selecting clones from one donor is critical to exclude studying interindividual differences rather than differences between individual

cellular lineages. Neither clustering nor PCA analysis of the data showed a conserved structure between preadipocytes and adipocytes that would allow us to group the cell lines into lineages. This suggests as also seen in the scRNAseq data that there are many subtle differences between the clonal cell lines, influenced by cell cycle state, proliferation capacity etc., which generate transcriptional noise complicating the assignment of individual cell lines into lineages.

Hence, we compared the RNAseq profile of each preadipocyte and adipocyte sample to the ProFat database to assess for each cell line its relative browning capacity. This comparison clearly demonstrated a separation of our cell lines, albeit all derived from the SVF of BAT, strongly indicating distinct subtypes of our clones. To identify potential lineage markers that we could use to visualize these clusters/subtypes in vivo, we used Laplacian eigenmaps to structure our data identifying seven distinct GEMs. From these genes, we selected those with conserved expression between preadipocytes and mature adipocytes and correlated each of them with the ProFat database. From these 57 genes, we selected EIF5, TCF25, and BIN1 as representatives for brown adipocytes with a classical brown adipocyte, intermediate, or more distant gene expression, respectively. We substantiated our in vitro findings through staining of mature brown adipocytes in BAT for EIF5, TCF25, or BIN1 revealing a contribution of each population to around 25% of brown adipocytes. Moreover, co-stainings with UCP1 confirmed the correlations observed in vitro.

It is important to note that in contrast to our previous work identifying white, beige and brown adipocyte specific surface proteins (16), neither of these genes are brown adipose selective. In contrast, all three genes show a broad tissue expression pattern. However, using different environmental challenges such as chronic cold exposure and HFD feeding, we find that *Eif5* and *Tcf25* expression closely parallel the expression of *Ucp1*, whereas *Bin1* expression was not regulated under each of these conditions in BAT in vivo (Fig 7K). In this context, it will be also very interesting to investigate the expression, regulation, and functional role of these genes in beige adipocytes within white fat depots.

The function of any of these genes in BAT, however, has not been studied yet. We find that loss of *Eif5* did not significantly impact on the brown adipocyte characteristics of these cells. Interestingly, loss of *Tcf25* shifted cellular identity to less brown-like adipocytes with up-regulation of *Asc1*, a white adipocyte marker, and reduction of maximum respiratory capacity to respire. On the other hand, *Bin1*$^{high}$ cells expressed low *Ucp1* and *Bin1* expression negatively correlated with thermogenic genes. Interestingly, loss of *Bin1* resulted in strongly increased *Ucp1* expression

shScr/shEif5/shTcf25/shBin1 preadipocytes. **(F)** Representative images of differentiated cell lines (shScr, shEif5, shTcf25, and shBin1) stained with F-actin (white), lipid droplets (red), and DAPI (blue). **(G)** Quantification of relative lipid accumulation measured by Oil Red O staining at day 8 of differentiation from knockdown cell lines (n = 4, mean OD normalized to DAPI ± SEM). **(H)** mRNA expressions of *Pparg* (left panel) and *Ucp1* (right panel) on shScr, shEif5, shTcf25, and shBin1 adipocytes and CL-316,243-treated adipocytes (0.5 μM CL-316,243 for 3 h; n = 6). **(I)** Quantification of Western blots for UCP1 (Cat. no. 10983; Abcam) in shScr, shEif5, shTcf25, and shBin1 adipocytes and adipocytes treated with 0.5 μM CL-316,243 for 6 h (n = 6–7) normalized to β-actin. Values are fold changes to the respective shScr adipocytes (ads). **(J)** Oxygen consumption rate of control and knockdown cell lines at day 8 of differentiation measured by Seahorse (n = 3); data were normalized by non-mitochondrial respiration. **(K)** Schematic illustration of brown adipose tissue heterogeneity: Murine brown adipose tissue is composed of functionally distinct brown adipocytes. We identify EIF5 expressing brown adipocytes as "classical" brown adipocytes, with high UCP1 content and mitochondrial uncoupling. TCF25 expressing brown adipocytes are similar, albeit with lower UCP1 expression. In contrast, BIN1-expressing brown adipocytes appear in a dormant state, expressing low UCP1 levels with little response to beta adrenergic stimulation. However, loss of BIN1 or chemical mitochondrial uncoupling reveals high thermogenic capacity of these cells. Subpopulations of brown adipocytes are color coded regarding to their thermogenic capacity, with the highest shown in dark brown (*Eif5*$^{high}$), followed by brown (*Tcf25*$^{high}$) and light brown (*Bin1*$^{high}$) (=indicates unchanged expression). All RT-qPCR data were normalized to *Tbp* and shown as mean ± SEM. *$P < 0.05$, **, ##$P < 0.01$, ***$P < 0.001$, ****$P < 0.0001$. Source data are available for this figure.

and mitochondrial activity, indicating that *Bin1* is not only a marker of "quiescent" brown adipocytes but actively suppresses the expression of *Ucp1* and mitochondrial respiration (Fig 7K).

We also show that the three markers *EIF5*, *TCF25*, and *BIN1* are present in human BAT, strongly supporting the notion that distinct brown adipocyte populations could also exist in humans. Yet, it is important to note that there may be several other markers to be identified that will help create an overall picture of interspecies brown adipocyte subtype determination and functional development. Moreover, future studies on adipose selective deletion of these putative lineage markers will need to address the specific function of these proteins in brown fat. In addition, genetic labeling of these cell populations will allow lineage-tracing experiments being essential to provide further evidence that *Eif5*, *Tcf25*, and *Bin1* indeed mark individual brown (pre-) adipocyte lineages, which is suggested but not proven by our data. Furthermore, the relative contribution and role of these subpopulations needs to be tested in other mouse strains.

Nevertheless, activation of BAT remains an attractive pharmacological goal to increase energy expenditure and combat the ever growing pandemic of obesity and the metabolic syndrome (46, 47, 48). However, anatomical, morphological, and potentially functional and developmental differences between rodent and human BAT complicate the translation from rodents to human physiology (26). At room temperature, morphology of murine BAT appears homogeneous, whereas human BAT appears as a mixture of unilocular and multilocular adipocytes (49, 50).

Conversely, at thermoneutrality or upon prolonged HFD feeding, murine BAT resembles the morphology of human BAT (50, 51). The appearance of unilocular adipocytes could either indicate the de novo differentiation of white adipocytes or excessive accumulation of triglycerides in brown adipocytes. The latter would suggest that individual brown adipocytes within murine BAT respond very differently to environmental changes, such as temperature and diet and this could indicate the existence of distinct brown adipocyte lineages with potentially different function. Moreover, if BATs were also composed of multiple brown adipocyte lineages in humans, differences in the relative contribution of these lineages could help explain differences between individuals with respect to BAT activity and the difficulties to translate findings from rodents housed at room temperature to humans.

In conclusion, we provide the first evidence for the existence of at least three functionally distinct brown adipocyte subtypes in mice that differ in their activation state (mitochondrial uncoupling and *Ucp1* expression) and provide a rich resource of data to extract and use this information for additional analysis. These data will foster translation by providing novel approaches to "humanize" murine BAT. Moreover, the presence of multiple human brown adipocyte subtypes and potential differences in the cellular composition between individuals or upon obesity and the metabolic syndrome could provide novel strategies for personalized obesity therapy.

# Materials and Methods

### Cell models and cell culture

Brown preadipocytes were isolated from interscapular BAT of three 8-wk-old, male WT C57BL/6 mice. BAT was minced and digested with 1 mg/ml type-A collagenase (Gibco) in DMEM (Gibco) for 30 min at 37°C. The SVF was isolated by centrifugation, plated on cell culture plates, and subsequently immortalized using an ecotropic retrovirus encoding the SV-40 large T antigen. Single-cell clones were picked, following a low-density plating of the immortalized preadipocytes and grown individually in separated plates.

Brown preadipocyte clones were cultured in normal growth medium (DMEM + GlutaMAX, 4.5 g/l D-glucose, pyruvate, 10% FBS, and 1% Pen-Strep). Differentiation of preadipocytes was induced with 0.5 mM IBMX, 5 $\mu$M dexamethasone, 125 $\mu$M indomethacin, 1 nM T3, and 100 nM human insulin, in growth medium for 2 d. Subsequently, medium was changed every 2 d with growth medium containing 1 nM T3 and 100 nM insulin, and the cells were harvested at day 8. For acute CL-316,243 treatment, the cells were stimulated with 0.5 $\mu$M CL-316,243 for 3 h and chronically with 0.1 $\mu$M CL-316,243 between days 2 and 8. Cell cultures were tested regularly negative for mycoplasma contamination.

Brown and subcutaneous preadipocytes were isolated and immortalized from an 8-wk-old male WT C57/BL6 mouse following the same protocol as above. Mature brown adipocytes were harvested at day 8 of differentiation as stated before. Subcutaneous preadipocytes were differentiated into white and beige adipocytes. For white adipocytes, subcutaneous preadipocytes were induced with 0.5 mM IBMX, 5 $\mu$M dexamethasone, and 100 nM human insulin in growth medium for 2 d. In addition, 1 $\mu$M rosiglitazone was supplemented to the same induction mix in growth medium for 2 d to differentiate subcutaneous preadipocytes into beige adipocytes. The medium was changed every 2 d with growth medium containing 100 nM insulin. Beige and white adipocytes were harvested at day 8 of differentiation.

For knockdown experiments, immortalized brown preadipocytes were infected with ecotropic lentiviral particles (Cell Biolabs, Inc.) with a scrambled (Scr), anti-Eif5 shRNA (TRCN0000313045: CCGGGGACGTTG-CAAAGGCGCTTAATCTCGAGATTAAGCGCCTTTGCAACGTCTTTTTG), anti-Tcf25 shRNA (TRCN0000241031: CCGGTGGAAAGAACCCGCCATTATGCTCGAGCA-TAATGGCGGGTTCTTTCCATTTTTG), or anti-Bin1 shRNA (TRCN0000238138: CCGGCCCGAGTGTGAAGAACCTTTCCTCGAGGAAAGGTTCTTCACACTCGGGTTTT TG). After 24 h, the cells were selected with growth medium containing 2.5 $\mu$g/ml puromycin (Biomol). The cells were maintained and cultured in growth medium with puromycin. CL-316,243 treatment on all knockdown experiments was performed by adding 0.5 $\mu$M CL-316,243 into growth medium containing 1 nM T3 and 100 nM insulin on day 8 of differentiation. Cells were harvested 6 h after the treatment, followed by RNA or protein extraction.

### Animal models

All animal studies were performed in a conventional animal facility with ad libitum access to food and water, 45–65% humidity, and a 12 h light–dark cycle. For the HFD study, wild-type male C57BL/6NCrl mice were purchased from Charles River at the age of 6 wk and maintained at constant ambient temperature of 22°C ± 2°C. The mice were fed low-fat containing control diet (Ssniff E15745-00). At the age of 7 wk, mice were either maintained on the control diet or were switched to a 60% kcal from fat HFD (Ssniff E15742-34) for 12 wk. The mice were euthanized with ketamine (100 mg/kg)/xylazine (7 mg/kg); the interscapular BAT was excised and frozen on dry ice.

Immunofluorescence pictures of Ucp1-KO BAT were taken from Ucp1-KO mice on a C57BL/6J (Jackson Laboratory) genetic background (strain name: B6.129-UCP1tm1Kz/J). The mice were bred, born, and weaned at 30°C. With 3 wk of age, the animals were transferred to room temperature and euthanized after 45 d.

For the cold exposure study, C57BL/6J mice were bred, born, and raised at 30°C. At the age of 10–12 wk, mice were single housed and randomly assigned to thermoneutral conditions (30°C) or cold (2–3 wk at 18°C followed by 4 wk at 5°C) acclimation. Afterwards, the mice were euthanized 3–4 h after lights went on, and BAT was collected. Animal experiments were conducted under permission and according to the German Animal Welfare Act and relevant guidelines and regulations of the government of Upper Bavaria.

## Lipid staining

Mature brown adipocytes were fixed in 10% formaldehyde for 1 h at room temperature and dehydrated with 60% isopropanol. Cells were stained with 60% ORO in water (stock: 0.35% ORO [Cat. no. O-0625; Sigma-Aldrich] in 100% isopropanol) for 10 min and subsequently washed six times with ddH$_2$O. DAPI staining was used for normalization, and the fluorescence signal was measured at 460 nm, using a microplate reader (PHERAstar FSX). To quantify the ORO content, ORO was eluted with 100% isopropanol and measured at 505 nm (PHERAstar FSX).

## The 3-(4,5-dimethylyhiazol-2-yl)-2,5-diphenyl-2H-tetrazolium bromide (MTT) assay

Preadipocyte clones were seeded in 96-well plates (2,000 cells/well). MTT stock solution was prepared by diluting 50 mg MTT powder (Serva) in 10-mL sterile PBS. On the day of measurement, 200 $\mu$l MTT stock solution was diluted in 1.6 ml DMEM and added to the wells. As the death control, 200 $\mu$l of DMEM medium containing 0.003% Triton X-100 and 0.05% MTT was added to the wells. After 2 h, the medium was removed and replaced by 100 $\mu$l solubilization buffer (10% Triton X-100 and 0.03% HCl in 100% isopropanol). The plate was incubated for 10 min at 24°C at 700 rpm using a ThermoMixer (Eppendorf). The lysate was then transferred to a new transparent 96-well plate and measured at 570 nm (PHERAstar FSX).

## RT-qPCR and (sc)RNAseq

RNA from mature brown adipocytes was extracted using the QuickExtract RNA extraction kit (Biozym), following the manufacturer's instructions. cDNA was synthesized by High-Capacity cDNA Reverse Transcription Kit (Applied Biosystem). Semiquantitative RT-qPCR was performed using iTaq Universal SYBR Green Supermix (Bio-Rad) in a CFX384 Touch Real-Time PCR Detection System (Bio-Rad). Relative mRNA expression was calculated after normalization to TATA-binding protein (*Tbp*) for cells, and *β*2-microglobulin (*B2m*) for tissues unless indicated otherwise in the figure legend.

For human gene expression analysis, tissue from six autopsies conducted at the NIH Clinical Center was collected from the following anatomic sites: superficial subcutaneous fat from the anterior abdomen (five donors), deep supraclavicular fat and periadrenal fat (six donors). Resected tissue was rinsed in PBS and placed immediately in

RNAlater (QIAGEN). RNA was extracted by homogenizing 100 $\mu$g tissue using an RNeasy Mini Kit (QIAGEN) according to the manufacturer's instructions. Total RNA concentration and purity were determined by spectrophotometer at 260 nm (NanoDrop 2000 UV-Vis Spectrophotometer; Thermo Fisher Scientific). RNA (1 $\mu$g) was converted to cDNA using the High-Capacity cDNA Reverse Transcription Kit (Applied Biosystems). Relative quantification of mRNA was performed with 3.5 $\mu$l cDNA used in an 11.5 $\mu$l PCR reaction for *ACTB*, *BIN1*, *TCF25*, *EIF5*, and *UCP1* using SYBR (Bio Basic) for *Bin1*, *Tcf25*, and *Eif5* and TaqMan Gene Expression Assay for *Ucp1*. Quantitative RT-PCR assays were run in duplicates and quantified in the ABI Prism 7900 sequence-detection system. All genes were normalized to the expression of the housekeeping gene *β-ACTIN*. Primers used are listed in Table S1.

For RNAseq, RNA was extracted with an RNeasy kit (QIAGEN), following the manufactures instructions. RNA Integrity Number values were determined using an automated electrophoresis (Agilent 2100 Bioanalyzer) and only RNAs with a RNA Integrity Number value >9 were used for further processing (Supplemental Data 2). Non-strand–specific, polyA-enriched RNA sequencing was performed as described earlier (52). Briefly, for library preparation, 1 $\mu$g of RNA was poly(A)-selected, fragmented, and reverse-transcribed with the Elute, Prime, Fragment Mix (Illumina). End repair, A-tailing, adaptor ligation, and library enrichment were performed as described in the Low Throughput protocol of the TruSeq RNA Sample Prep Guide (Illumina). RNA libraries were assessed for quality and quantity with the Agilent 2100 BioAnalyzer and the Quant-iT PicoGreen dsDNA Assay Kit (Life Technologies). RNA libraries were sequenced as 100-bp paired-end runs on an Illumina HiSeq4000 platform. The STAR aligner (53) (v 2.4.2a) with modified parameter settings (–twopassMode=Basic) is used for split-read alignment against the human genome assembly mm9 (NCBI37) and UCSC knownGene annotation. To quantify the number of reads mapping to annotated genes we use HTseq-count (54) (v0.6.0). FPKM (Fragments Per Kilobase of transcript per Million fragments mapped) values are calculated using custom scripts. A sample overview is provided in Table S4.

### *scRNAseq*

Single-cell libraries were generated using the ChromiumTM Single cell 3' library and gel bead kit v2 (PN #120237) from 10x Genomics. Briefly, live cells from the SVF of BAT of 8-wk-old mice were obtained by flow cytometry following dead cell exclusion using 7-AAD. Afterwards, the cells were loaded onto a channel of the 10× chip to produce Gel Bead-in-Emulsions (GEMs). This underwent reverse transcription to barcode RNA before cleanup and cDNA amplification followed by enzymatic fragmentation and 5' adaptor and sample index attachment. Libraries were sequenced on the HiSeq4000 (Illumina) with 150-bp paired-end sequencing of read 2 and 50,000 reads per cell.

## Western blot

Mature brown adipocytes were lysed with radioimmunoprecipitation assay buffer (RIPA buffer) (50 mM Tris, pH 7.4, 150 mM NaCl, 1 mM EDTA, and 1% Triton X-100), containing 0.1% SDS, 0.01% protease-inhibitor, 0.01% phosphatase-inhibitor cocktail II, and 0.01% phosphatase-inhibitor

cocktail III (all from Sigma-Aldrich). Protein concentrations were measured using a BCA Protein Assay Kit (Thermo Fisher Scientific), with BSA dilution series as the standard. Proteins were separated by SDS–PAGE, with Fisher BioReagents EZ-Run Prestained *Rec* Protein Ladder (Thermo Fisher Scientific), as the molecular weight marker and transferred to a polyvinylidene fluoride (PVDF) Immobilon-P^SQ membrane, 0.2 μm (Merck Millipore). Unspecific binding sites were blocked with 5% BSA/TBS-T. The membranes were incubated with primary antibody solutions, Abcam (Cat. no. ab10983), 1:1,000. The membranes were washed three times (each 10 min), with 1× PBS before incubation with secondary HRP-conjugated antibody (Cat. no. 7074, 1:10,000; Cell Signaling Technology). β-actin (HRP-linked, Cat. no. sc-47778, 1:5,000; Santa Cruz Biotechnology) was used as loading control. Quantifications were performed using Image J software.

### Immunostainings

Brown adipocyte clones were cultured on 96-well glass-bottom plates, coated with 0.1% gelatin (Cat. no. G1890; Sigma-Aldrich). Preadipocytes as well as fully differentiated brown adipocytes (day 8) were fixed with 10% formaldehyde and blocked with 3% BSA/PBS. Alexa Fluor 546 Phalloidin (Cat. no. A22283, 150 nM; Invitrogen), HCS LipidTOX Green Neutral Lipid Stain (Cat. no. H34775, 1:200; Invitrogen), and DAPI (1 μg/ml; Sigma-Aldrich), were used to stain F-actin, lipids and nuclei, respectively. The cells were imaged using the Operetta High-Content Imaging System (20× magnification; PerkinElmer).

Interscapular BAT was collected from 3- to 4-mo-old, male C57BL/6J WT mice and fixed in 4% paraformaldehyde in PBS for 30 min at room temperature on a shaker. BAT was embedded in 4% low melting temperature agarose (A9414; Sigma-Aldrich) in PBS and cut into 70-μm sections using a Leica VT1000 S Vibrating blade microtome. Sections were permeabilized with 1% Triton X-100/PBS on ice for 1 min and blocked with 3% BSA/PBS for 1 h. Primary antibodies were added for detection of UCP1 (Cat. no. ab10983, 1:250; Abcam), EIF5 (Cat. no. ab170915, 1:100; Abcam), TCF25 (Cat. no. PA521418, 1:200; Invitrogen), and BIN1 (Cat. no. ab182562, 1:200; Abcam), followed by Alexa Fluor 594 antirabbit (Cat. no. ab150080, 1:400; Abcam) as secondary antibody, Alexa Fluor 647 Phalloidin (Cat. no. A22287, 1:100; Invitrogen), HCS LipidTOX Green Neutral Lipid Stain (Cat. no. H34775, 1:200; Invitrogen), and DAPI (1:1,000; Sigma-Aldrich). Stained sections were mounted on microscope slides using DAKO fluorescence mounting medium (Cat. no. S3023; Agilent). Images were captured using a Leica TCS SP5 Confocal microscope (Leica Microsystems).

Co-stainings of TCF25, EIF5, and BIN1 with UCP1 were carried out as described above, but after the secondary antibody staining, the sections were incubated with Alexa Fluor 488–conjugated UCP1 antibody (Cat. no. ab225490, 1:100; Abcam), Alexa Fluor 647 Phalloidin (Cat. no. A22287, 1:100; Invitrogen), and DAPI (1:1,000; Sigma-Aldrich) for 2 h at room temperature. Stained sections were mounted on microscope slides using DAKO fluorescence mounting medium (Cat. no. S3023; Agilent). Images were captured using a Leica TCS SP5 Confocal microscope (Leica Microsystems).

Automated quantifications of UCP1 and EIF5 were performed using the commercially available image analysis software Definiens Developer XD 2 (Definiens AG) following a previously published procedure (55). A specific rule set was developed to detect and quantify UCP1 and EIF5 stained tissue: In a first step the images were segmented based on color and shape features. The calculated parameter was the mean staining intensity of UCP1 or EIF5 in the detected cells, and they were divided in four classes using the fluorescent intensity of each marker.

### Seahorse analysis

Cells were plated in 24- or 96 (for knockdown cells)-well Seahorse plates and differentiated as described above. The OCR was measured at day 8 of differentiation using a XF24 or XF96 Extracellular Flux analyzer (Seahorse). 1 h before measurement, the cells were equilibrated at 37°C, in assay medium (DMEM D5030 supplemented with 0.2% fatty acid-free BSA, 25 mM glucose, 1 mM sodium pyruvate, and 4 mM L-Glutamine [Sigma-Aldrich]). Compounds were diluted in assay medium and loaded to the equilibrated cartridge ports (A: 20 μM isoproterenol, B: 15 μM oligomycin, C: 250 μM DNP, D: 40 μM antimycin A, and 25 μM Rotenone) and calibrated. Measurements were taken before and after each injection through four cycles, with each cycle comprising 3 min mixing, 2 min waiting, and 2 min measuring. Data were normalized with non-mitochondrial respiration. ATP-linked OCR value was calculated by subtracting basal OCR from proton leak OCR.

### Computational analysis of the scRNA-seq data

For the analysis, an indexed mm10 reference genome was build based on the GRCm38 assembly and genome annotation release 94 from Ensembl. For the alignment, quality control, the estimation of valid barcodes, and creating the count matrices, the Cell Ranger pipeline (version 2.1.1, from 10x genomics, https://support.10xgenomics.com) was run with the command "cellranger count" with standard parameters, except that the number of expected cells was set to 10,000, the chemistry was set to "SC3Pv2." An anndata object was created using the python package Scanpy (33) (version 1.0.4). The full analysis is shown in Supplemental Data 1.

#### Cell type assignment
The workflow was performed in scanpy version 1.4.3 (56). The cells were filtered based on a unique molecular identifier counts and the fraction of mitochondrial RNA. The remaining cell vectors were normalized to sum a total count of $1 \times 10^4$ by linear scaling, log(x + 1)-transformed, and highly variable genes were selected. We then performed PCA with 50 principal components (PC) and used the PC space to compute a k-nearest neighbor (kNN) graph (k = 100, method = umap). We computed uniform manifold approximation and projection (UMAP) and a louvain clustering (resolution = 1, flavor = vtraag) based on the kNN graph (louvain_1). We assigned cell types to clusters based on marker gene expression by cluster.

#### Heterogeneity analysis of preadipocytes
We selected the louvain clusters that correspond to preadipocytes and putative mature adipocytes from the overall clustering (louvain_1) and separately processed these cells: normalizing to $1 \times 10^4$ counts, then log(x + 1) transforming the data and selected highly

variable genes followed by PCA with 50 PCs. Again, we computed a kNN graph (k = 100, method = umap) based on the PC space. We computed UMAP and a louvain clustering (resolution = 1, flavor = vtraag) based on the kNN graph; we call this clustering louvain_2 in the following.

### Statistical analysis

Data are shown as mean ± SEM, if not indicated otherwise in the figure legend. Statistical significance for multiple comparisons was determined by one- or two-way ANOVA, with Tukey's multiple comparisons test, or unpaired $t$ test (two-tailed $P$-value). Correlation graphs were analyzed with linear regression (two-tailed $P$-value, 95% CI). GraphPad Prism 6 was used for statistical analysis. $P$-values < 0.05 were considered as statistically significant.

RNA count files were normalized using R package DESeq2. Genes where all expression values were in the lowest 25% of the data were removed. In addition, genes with a variance in the lowest 25% were removed from the data. Hierarchical clustering was performed using "Euclidean" distance measure and nearest distance linkage method, if not indicated otherwise.

Nonlinear dimensionality reduction was performed using Laplacian eigenmaps (57) with Mahalanobis distance as similarity measure and $k$ = 23 for nearest neighbor graph generation. First eight eigenvectors of the graph Laplacian matrix were used as a low dimensional representation of preadipocyte expression matrix. For each gene in eigenvector-space we calculated the standardized Euclidean distance to geometrical median. K means with $k$ = 2 was used to remove "uninformative" proximal genes from the data. Silhouette was used to estimate an optimal $k$ = 7 for k-means clustering of the remaining distal genes. KEGG pathway enrichments were calculated using hypergeometrical distribution tests. Network plots were created using Cytoscape v3.6. Calculations were done using R 3.4 and MATLAB 2017b.

# Data Availability

RNAseq data are available at Gene Expression Omnibus: Series GSE122780. scRNAseq are available at Gene Expression Omnibus: GSE161447.

# Supplementary Information

# Acknowledgements

This work was supported by the project Aging and Metabolic Programming (AMPro) to S Ussar and by the European Research Council ERC Starting Grant (AstroNeuroCrosstalk no. 757393) to C García-Cáceres.

## Author Contributions

R Karlina: conceptualization, data curation, formal analysis, validation, investigation, visualization, methodology, and writing—original draft, review, and editing.
D Lutter: conceptualization, data curation, software, formal analysis, validation, visualization, methodology, and writing—original draft, review, and editing.
V Miok: data curation, software, formal analysis, validation, and visualization.
D Fischer: data curation, software, formal analysis, and writing—original draft.
I Altun: data curation, formal analysis, and investigation.
T Schöttl: data curation, formal analysis, and investigation.
K Schorpp: data curation, formal analysis, and investigation.
A Israel: formal analysis, investigation, and methodology.
C Cero: formal analysis, investigation, and methodology.
JW Johnson: formal analysis, investigation, and methodology.
I Kapser-Fischer: formal analysis, investigation, and methodology.
A Böttcher: formal analysis, investigation, and methodology.
S Keipert: formal analysis, investigation, and methodology.
A Feuchtinger: formal analysis, investigation, visualization, and methodology.
E Graf: investigation and methodology.
T Strom: methodology.
A Walch: methodology.
H Lickert: conceptualization and methodology.
T Walzthoeni: data curation, formal analysis, and methodology.
M Heinig: data curation, formal analysis, and methodology.
FJ Theis: conceptualization.
C García-Cáceres: conceptualization.
AM Cypess: conceptualization, data curation, formal analysis, investigation, methodology, and writing—original draft.
S Ussar: conceptualization, data curation, formal analysis, supervision, funding acquisition, visualization, methodology, project administration, and writing—original draft, review, and editing.

## Conflict of Interest Statement

The authors declare that they have no conflict of interest.

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
