## [Reviewer comments · Life Science Alliance]

Life Science Alliance

Identification and characterization of distinct brown adipocyte subtypes in C57BL/6J mice

Siegfried Ussar, Ruth Karlina, Dominik Lutter, Viktorian Miok, David Fischer, Irem Altun, Theresa Schoettl, Kenji Schorpp, Andreas Israel, Cheryl Cero, James Johnson, Ingrid Kapser-Fischer, Anika Boettcher, Susanne Keipert, Annette Feuchtinger, Elisabeth Graf, Tim Strom, Axel Walch, Heiko Lickert, Thomas Walzthoeni, Matthias Heinig, Fabian Theis, Cristina Garcia-Caceres, and Aaron Cypess

DOI: <https://doi.org/10.26508/lisa.202000924>

Corresponding author(s): *Siegfried Ussar, Helmholtz Diabetes Center*

Review Timeline:

Submission Date:	2020-09-30
Editorial Decision:	2020-09-30
Revision Received:	2020-11-06
Editorial Decision:	2020-11-11
Revision Received:	2020-11-13
Accepted:	2020-11-15

Scientific Editor: Shachi Bhatt

Transaction Report:

Please note that the manuscript was previously reviewed at another journal and the reports were taken into account in the decision-making process at Life Science Alliance.

Referee #1 Review

Report for Author:

Comments to the manuscript:

The manuscript entitled "Identification and characterisation of distinct murine brown adipocyte lineages" by Karlina et al. investigates the heterogeneity within mouse brown adipocytes using single cell RNAseq analysis of the stroma vascular fraction of murine BAT and different brown preadipocyte clones. The authors perform several studies to separate these populations and identify three markers: Eif5, Tcf25 and Bin1. Functionally, the authors showed that Bin1 depletion led to an increase in Ucp1 expression and mitochondrial respiration, suggesting this gene to be a marker of a dormant brown adipocyte type.

The novelty of the manuscript is the information about the heterogeneity of mouse brown adipocyte populations and the identification of new markers characterising different subpopulations of these cells *in vivo*. These findings indicate potentially distinct functions in thermogenesis and regulation of body energy homeostasis. The heterogeneity of the subpopulations of murine BAT have already been described by Song et al. (PMID: 31573981) and somehow by Sun et al (BioRxiv, doi: <https://doi.org/10.1101/2020.01.20.890327>).

The rationale for this work is interesting and important for the brown adipose tissue field. However, we feel that the authors do not provide enough validations of their findings. To be sure that what they see is not dependent on the genetic background of the mice, they should repeat their analysis on at least another mouse strain. Moreover, for the in-depth transcriptional analysis of their clones, the authors focus on cell lines coming only from one mouse (BAT1). They should also provide the same kind of analysis for the cell lines generated from the other two mice (BAT2 and 3), to validate their results. Further characterisation of the original clones and the knockdown cell lines should also be added.

Major comments:

- The authors use C57BL/6J mice for this study. Would they obtain the same results with other strains of mice, for example, C57BL/6N or 129svj? Or are these findings mouse strain-specific? The authors should validate their findings in another strain of mice, to confirm that what they found is valid for murine models in general and not genetic background dependent.
- The number of brown adipocyte markers used in most of the experiments is limited (mainly Pparg, Ucp1 and Prdm16). The authors should use a broader range of BAT genes to characterise both the BAT populations and the phenotype of the knockdown cell lines (Ebf2, Cebpb, Dio2, Pparg2, Cidea, Zic1...).
- To do more detailed analyses of the transcriptional differences between cell lines, the authors focus their work on cell lines coming from a single mouse (supplemental Fig.3 and Fig.4 onwards). However, they should repeat the analysis also on the cell lines coming from the other two mice to validate their findings.
- Supplementary Figure 1B and Figure 1B. Which method did the authors use to quantify the levels of UCP1 expression here?
- Figure 1C and Supplementary data1. Which other markers apart from Pdgfra were used to identify the two brown preadipocyte populations? The authors should indicate them in the text.
- Figure 2. Would the immortalisation of the brown adipocytes used to generate the 67 clonal cell lines not alter the properties and the gene expression of these brown adipocyte vs primary brown adipocytes? Sometimes the immortalisation method can change the nature of the cells.
- Figure 3B and supplementary Figure 6A. There is a debate in considering CD137 a beige adipocyte marker. The paper by Srivastava et al., PMID: 31919095, identifies CD137 as a negative regulator of browning. The authors should select another beige adipocyte marker for their analysis together with Tmem26.
- Supplementary figure 3 should be included in the leading figures since there is a whole section dedicated to it. The data presented in this figure are reproducible also for the clones coming from the other two mice (BAT1 and BAT2)? The authors should show the reproducibility of their analysis.
- Figure 4C is difficult to understand; a more detailed explanation should be added in the text. Moreover, the axis titles are missing.
- In page 17, the authors claim that "Bin1 marks a relatively "white adipocyte like" brown adipocyte population". Given that the experiments investigating the function of this gene in brown adipocytes are merely oxygen consumption measurements and Ucp1 expression levels, it would be more appropriate to describe the Bin1 high population as less thermogenic instead of "white adipocyte-like".

- According to the dendrogram in figure 3B (although, again, based on a very limited number of markers), some of the clones are more similar to beige or white adipocytes. The clones selected for the knockdown of Eif5 and Tcf25 clusters with beige adipocytes and the one used for Bin1 clusters with white adipocytes. Have the authors checked the expression of these markers in beige/white fat upon cold exposure or adrenergic treatment?
- Figure 5D-E. How do the authors explain the fact that Bin1 mRNA expression seems not regulated by cold exposure or HFD, but its depletion is increasing the expression of Ucp1 and mitochondrial uncoupling?
- Figure 6H. Why is there a reduction of Pparg expression in the Bin1 kd cells treated with CL?
- Supplementary Figure 6A. How do the authors explain the decrease in Ucp1 and Prdm16 expression in the Tcf25 kd cells vs an increase in Pgc1a expression and no difference in mitochondrial uncoupling in the Tcf25 kd cells (Figure 6C)?
- The results related to the Eif5 knockdown cell lines are not clear. According to Figure 6, shEif5 cells have higher expression of Ucp1 (Fig. 6H, I, Fig. S6A) and the brown fat marker P2rx5 (Fig. S6A). However, according to Figure 5 the Eif5high population is associated to high Ucp1 and P2rx5 expression. The authors should elaborate on this. What would be expected upon Eif5 overexpression?
- The manuscript would be easier to read if information in the discussion was also included in the respective sections. This applies especially to the known functions of the selected genes and a more detailed explanation of the use of the ProFat database.

Minor comments:

- More attention should be paid to nomenclature. There are multiple mistakes through the manuscript and the figures concerning the use of capitals, non-capitals and italics when talking about genes (i.e. page 4 'Ucp1 expression' should be 'Ucp1 expression', page 17 'P2XR5' should be 'P2xr5').
- Figures 1E and 4D are difficult to read; please use lighter colours.
- There are several typos and missing words in the manuscript; please correct them.

Referee #2 Review

Report for Author:

In the current study, Karlina, Lutter et al. aimed to investigate adipocyte heterogeneity of murine brown adipose tissue. While the functional and developmental heterogeneity of adipocytes is emerging as an important feature in white adipose depots and human thermogenic fat, it is unknown whether different subpopulations of brown adipocytes coexist in classical murine brown fat depots. By using whole SVF single cell sequencing, cell population cloning, and computational approaches, the authors aimed at distinguishing different brown adipocyte subpopulations. While the authors found that in vivo differences primarily relate to different developmental stages, careful analysis of clonal cell-lines, which were differentiated in vitro, revealed the presence of at least 3 different subsets. The authors further identified three markers for these different classes of brown adipocytes in vitro, and detected their presence in BAT in vivo. Finally, they characterized the functional relevance of these markers in modulating the thermogenic potential of these cells. Overall, this elegant study brings important new information about the identity of distinct precursor cells for energy-dissipating brown adipocytes in mice, with potential implications in our understanding of the regulation of energy homeostasis. Addressing the following comments would further support the conclusions raised in this study.

Major comments:

- In Figures 2 and 3, the authors describe the overall heterogeneity of the 67 cell-lines they obtained, by assessing lipid accumulation and expression of key adipogenic genes such as Pparg, Ucp1, and few additional brown, beige and white markers, revealing quite a large disparity in the differentiation potential of these cells (e.g. Fig. 2A, Fig. 2D, Fig 3A). It would therefore be informative to provide a more comprehensive description of the intrinsic differences of the different clones, to properly interpret the data. This could be addressed by evaluating the expression of additional key genes for adipocyte functions (such as Fabp4, Cd36, Fasn, AdipoQ, Lipe, or others involved in glucose and lipid metabolism, insulin action, and mitochondrial function).
- Figure 5F and 5G describes the cellular distribution of Eif5, Tcf25 and Bin1 in BAT by immunostaining. However, to investigate how these markers associate with different subcategories of brown adipocytes in vivo (Ucp1 low to high, Fig. 1 A-B), it would be necessary to perform co-staining with Ucp1.
- The authors evaluated the potential functional significance of the 3 selected markers in brown adipocytes following stable knockdown in clonal brown adipocyte cell-lines or in a mixed population. Whereas Eif5 is associated with higher thermogenic potential (Fig 5), Eif5 knockdown leads to increased Ucp1 expression (Supp Fig 6A, Fig 6H), without any change in mitochondrial respiration (Fig 6C and J). On the contrary, Bin1 is associated with lower thermogenic potential (Fig. 5), and Bin1 knockdown leads to higher Ucp1 expression (Supp Fig 6A, Fig 6H), and increased mitochondrial respiration (Fig 6C and J). While these data strongly suggest a direct involvement for the selected markers, especially Bin1, in regulating brown adipocyte functions, some interrogations remain. In particular, it is puzzling that the knockdown of Eif5 or Bin1, marking opposite subpopulations, leads to similar effects at the gene expression level, with a different functional outcome. To further evaluate whether the role of Eif5, Tcf25 and Bin1 in regulating the thermogenic potential of brown adipocytes is cell-intrinsic or rather results from a more complex interplay between different brown adipocyte subpopulations (Fig 6D-J), it would be helpful to test the effect of Eif5, Tcf25 and Bin1 knockdown in different representative (Eif5^{high}, Tcf25^{high}, Bin1^{high}) clonal cell-lines.

Minor comments:

- The claim that "loss of Bin1 results in increased thermogenic gene expression and full mitochondrial uncoupling, at least in clones with high Bin1 expression" is overstated at this point since Ucp1 was the only thermogenic gene reported to be regulated, and a unique clonal cell-line was investigated. The authors should provide further data, or need to adjust their conclusion.
- It would be interesting to know what the authors speculate about the function of these different populations in BAT biology.

Referee #3 Review

Report for Author:

This new study from Ussar and colleagues set out to explore the hypothesis of distinct brown adipocyte lineages within murine brown adipose tissue. The authors utilize scRNA-seq, clonal analyses of immortalized brown stromal cells, and computational analysis. They conclude that multiple brown adipocytes lineages are present in vivo, with distinct functional properties.

Overall, the paper addresses an important and fundamental question in adipose developmental biology. However, the data and approach are not adequate to support the conclusions presented.

The main conclusion lies in the notion that there are "functionally" and "developmentally distinct" brown adipocyte lineages. First and foremost, there is very little functional analysis of identified cell populations emanating from the scRNA-seq analysis or the clonal analyses of immortalized cells. Moreover, the notion that the identified clonal lines represent natural adipocytes of distinct lineages in vivo is not supported by any lineage tracing or classical developmental analyses.

Specific points:

1) The authors conclude that the populations identified from the scRNA-seq analysis represent different stages of the differentiation process, rather than distinct lineages. This argument is based solely on computational analyses without any attempts to isolate the populations and study them functionally. The authors should refer to the Merrick et al. Science 2019 paper on how this can be achieved.

2) Two things about the derived clonal cell lines are not convincing. A) Whether these distinct cell lines reflect cells that are naturally present in vivo, and B) whether there are truly functional differences between the cell lines.

The authors demonstrate that cell lines differ in their ability to activate Ucp1 mRNA. Does this translate into functional differences in thermogenesis?

The shRNA experiments nicely point to a functional role for Eif5, Tcf25, and Bin1, but do Eif5High, Tcf25high, and Bin1High cells actually differ functionally? This needs to be evaluated.

3) Most importantly, the notion of these cell populations representing distinct lineages is not well supported. Eif5High, Tcf25high, and Bin1High cells in vivo may simply reflect cells of distinct "states" rather than developmental lineages. Lineage tracing is needed to support this idea.

September 30, 2020

Re: Life Science Alliance manuscript #LSA-2020-00924-T

Dr. Siegfried Ussar
Helmholtz Diabetes Center
JRG Adipocytes & Metabolism
Ingolstädter Landstraße 1
GAR/01.03.51
Neuherberg D-85764
Germany

Dear Dr. Ussar,

Thank you for submitting your manuscript entitled "Identification and characterization of distinct murine brown adipocyte lineages" to Life Science Alliance. The manuscript was assessed by expert reviewers, whose comments are appended to this letter.

For a brief overview, the manuscript was peer-reviewed at another journal previously. The reviewers were concerned about the advance, given the prior literature, and a lack of sufficient functional in vivo data for the newly identified three BAT sub-populations and their markers. After a detailed assessment of the manuscript and reviewers comments, Life Science Alliance (LSA) editors deemed that the study can be published at LSA with minor edits, and toning down some of the conclusions made. The following edits are required for re-submission to LSA -

+ Please provide a point-by-point rebuttal to the reviewers comments

+ Textual changes / Discussion / Clarity

++ Rev 1's points 1-4, 8-10, 13-14, and their minor comments should be discussed and clarified explicitly

++ Rev 3 pt 3 manuscript text must be edited accordingly

Since the paper does not quite define the functionality of these sub-cell types and is mostly based on analysis on one mouse strain, and clonal cell lines from only one mouse - these points should be reflected in the title, and main message and apparent in the abstract as well

+ Experimental requests

++ Rev 1 pt 5, 15; and Rev 2 pt 1-3 should be addressed with experiments in the revised manuscript

+ Good to have, but not required for publication at LSA

++ Rev 1 pt 6 - please add new beige cell markers in the analysis if possible, otherwise please clarify the use of CD137 as a beige cell marker, and the concerns pertaining to that. How would these concerns affect interpretation of your results

++ Rev 1 pt 11 - If possible and easily attainable, we would encourage you to include data from expression of Eif5, Tcf25 and Bin1 in beige/white fat after cold exposure or adrenergic treatment. If not, we would appreciate some speculation of what the effects might be

++ Rev 3 pt 2b - If possible, please include data showing the effects of Eif5High, Tcf25high, and

Bin1High cells actually differ functionally, particularly if there is any difference in thermogenesis between these 'sub-populations'- if not please tone down the statement suggesting these to be different lineages or sub-populations in lieu of lacking functional data

We understand that these revisions might take some time, and would require re-review. The LSA editors will, of course, explain the entire transfer process to the reviewer(s) when we send them the revised study.' We would be happy to discuss the individual revision points, over email or phone, should this be helpful.

While you are revising your manuscript, please also attend to the below editorial points to help expedite the publication of your manuscript. Please direct any editorial questions to the journal office. The typical timeframe for revisions is three months. Please note that papers are generally considered through only one revision cycle, so strong support from the referees on the revised version is needed for acceptance.

Thank you for considering Life Science Alliance as an appropriate venue for your research. We look forward to receiving your revised manuscript.

Sincerely,

Shachi Bhatt, Ph.D.
Executive Editor
Life Science Alliance

- A letter addressing the reviewers' comments point by point.
- An editable version of the final text (.DOC or .DOCX) is needed for copyediting (no PDFs).
- High-resolution figure, supplementary figure and video files uploaded as individual files: See our detailed guidelines for preparing your production-ready images, <https://www.life-science-alliance.org/authors>
- Summary blurb (enter in submission system): A short text summarizing in a single sentence the study (max. 200 characters including spaces). This text is used in conjunction with the titles of papers, hence should be informative and complementary to the title and running title. It should describe the context and significance of the findings for a general readership; it should be written in the present tense and refer to the work in the third person. Author names should not be mentioned.

B. MANUSCRIPT ORGANIZATION AND FORMATTING:

Editorial comment: Since the paper does not quite define the functionality of these sub-cell types and is mostly based on analysis on one mouse strain, and clonal cell lines from only one mouse - these points should be reflected in the title, and main message and apparent in the abstract as well

We modified the title, abstract and text accordingly.

Referee #1 ++ Rev 1's points 1-4, 8-10, 13-14, and their minor comments should be discussed and clarified explicitly

Major comments:

Point 1: The authors use C57BL/6J mice for this study. Would they obtain the same results with other strains of mice, for example, C57BL/6N or 129svj? Or are these findings mouse strain-specific? The authors should validate their findings in another strain of mice, to confirm that what they found is valid for murine models in general and not genetic background dependent.

We focused our analysis on C57BL/6J as the most commonly used mouse model for metabolic studies. We understand the limitations of using only one mouse strain in our study. However, we would like to point out that in the overwhelming amount of mouse studies only one mouse strain is used. It is beyond our financial possibilities to repeat all experiments in our manuscript with different mouse strains. To this end we modified the title to reflect this limitation more clearly and also discuss this point in the manuscript.

Point 2: The number of brown adipocyte markers used in most of the experiments is limited (mainly Pparg, Ucp1 and Prdm16). The authors should use a broader range of BAT genes to characterise both the BAT populations and the phenotype of the knockdown cell lines (Ebf2, Cebpb, Dio2, Pparg2, Cidea, Zic1...).

The main aim of our manuscript is to describe heterogeneity of murine brown adipose tissue and to provide a resource to study the mechanistic underpinnings of this heterogeneity. To this end, we focused our analysis on a limited number of well-established markers for brown adipocyte function and adipose tissue in general. However, we would like to point out that all our single cell RNAseq and RNAseq data will be publically available and expression of any given marker can be extracted from these data. We now encourage the reader to utilize this resource.

We added expression data on several additional markers (*Adiponectin*, *Cd36*, *Fabp4*, *Fasn*, *Glut4* and *Hsl*) in BAT clones (Supplementary Figure 2B).

Point 3: To do more detailed analyses of the transcriptional differences between cell lines, the authors focus their work on cell lines coming from a single mouse (supplemental Fig.3 and Fig.4 onwards). However, they should repeat the analysis also on the cell lines coming from the other two mice to validate their findings.

We focused on the cell lines from one mouse to overcome mouse-to-mouse differences and truly study differences of cells from one single tissue. We performed RNAseq from undifferentiated and differentiated BAT1 clones. However, we do not have the financial abilities to perform additional 94 RNAseq runs. However, we validated the expression of the three marker genes *in vivo* using different mice.

Point 4: Supplementary Figure 1B and Figure 1B. Which method did the authors use to quantify the levels of UCP1 expression here?

As stated in the materials and methods; the levels of UCP1 were quantified by the image analysis software Definiens Developer XD 2 using a published method [DOI: 10.1007/s00418-014-1258-2]. Based on algorithms, the software is able to detect and quantify the staining intensities in different cellular compartments.

Point 5: Figure 1C and Supplementary data1. Which other markers apart from *Pdgfra* were used to identify the two brown preadipocyte populations? The authors should indicate them in the text.

In addition to *Pdgfra* we used *Fgf10*, *Zfp423* and *Cd34*, as it is shown in Supplementary Data 1.

Point 6: Figure 2. Would the immortalisation of the brown adipocytes used to generate the 67 clonal cell lines not alter the properties and the gene expression of these brown adipocyte vs primary brown adipocytes? Sometimes the immortalisation method can change the nature of the cells.

We fully share the concern of the reviewer, which is why we include data on cellular proliferation rates, as a surrogate for the effects of SV40 large T expression, between clones, where we could not detect significant differences. Unfortunately, it would be impossible to perform the here described experiments and clonal isolation with primary cells due to senescence. However, we would like to emphasize that we validated our identified markers *in vivo* by immunofluorescence stainings in mature adipocytes and scRNAseq in precursors.

Point 7 please add new beige cell markers in the analysis if possible, otherwise please clarify the use of CD137 as a beige cell marker, and the concerns pertaining to that. How would these concerns affect interpretation of your results: Figure 3B and supplementary Figure 6A. There is a debate in considering CD137 a beige adipocyte marker. The paper by Srivastava et al., PMID: 31919095, identifies CD137 as a negative regulator of browning. The authors should select another beige adipocyte marker for their analysis together with *Tmem26*.

We appreciate the reviewer bringing up this important point, as we observe a similar situation with EIF5, where EIF5 marks a UCP1 high subpopulation of brown adipocytes, yet, knockdown of *Eif5* enhances thermogenic gene expression.

However, we do not understand why the reviewer suggests to use other beige adipocyte markers? Srivastava do not question the validity of CD137 as marker for beige adipocytes, but rather question is use as a functional marker, implying that increased CD137 expression within a single cells correlates with increased thermogenic capacity. Here we focus on the mRNA expression levels of CD137, which are widely used to classify beige adipocytes and not protein levels. Thus, we do not see any conflicting data on using CD137 mRNA levels as marker for beige adipocytes.

Point 8: Supplementary figure 3 should be included in the leading figures since there is a whole section dedicated to it. The data presented in this figure are reproducible

also for the clones coming from the other two mice (BAT1 and BAT2)? The authors should show the reproducibility of their analysis.

We initially moved this figure to the supplement due to limitations in the number of main figures. We have now moved the data to main Figure 3. As outlined above we did not do the RNAseq experiments for the clones from the other two mice but confirmed our results in vivo using immunohistochemistry.

Point 9: Figure 4C is difficult to understand; a more detailed explanation should be added in the text. Moreover, the axis titles are missing.

We apologize for the missing axis. We added the axis title for Figure 4C and provide additional explanation for the interpretation of the results.

Point 10: In page 17, the authors claim that "Bin1 marks a relatively "white adipocyte like" brown adipocyte population". Given that the experiments investigating the function of this gene in brown adipocytes are merely oxygen consumption measurements and Ucp1 expression levels, it would be more appropriate to describe the Bin1 high population as less thermogenic instead of "white adipocyte-like".

We changed the text according to the reviewer's suggestion.

Point 11 Rev 1 pt 11 - If possible and easily attainable, we would encourage you to include data from expression of Eif5, Tcf25 and Bin1 in beige/white fat after cold exposure or adrenergic treatment. If not, we would appreciate some speculation of what the effects might be. According to the dendrogram in figure 3B (although, again, based on a very limited number of markers), some of the clones are more similar to beige or white adipocytes. The clones selected for the knockdown of Eif5 and Tcf25 clusters with beige adipocytes and the one used for Bin1 clusters with white adipocytes. Have the authors checked the expression of these markers in beige/white fat upon cold exposure or adrenergic treatment?

All three genes are expressed in multiple tissues including white adipose tissue. The focus of the current study was to study heterogeneity within murine interscapular brown adipose tissue. Studying beige adipocytes will certainly be of great interest in the future. However given the already existing confusion/discussion on the differences between beige and brown adipocytes and mice and humans, we would like to avoid any confusion of the reader by including data on beige adipocytes in this manuscript, as we focus solely on brown adipose tissue.

Point 12: Figure 5D-E. How do the authors explain the fact that Bin1 mRNA expression seems not regulated by cold exposure or HFD, but its depletion is increasing the expression of Ucp1 and mitochondrial uncoupling?

We understand that this observation could cause confusion. However, we believe that it is still important to report these findings. There are several potential explanations for this observation. Bin1 is also highly expressed in endothelial cells, and thus, using whole tissue lysates, changes in the expression of Bin1 in brown adipocytes could be masked by the endothelial expression. Alternatively, Bin1 positive cells could be unresponsive to these changes in diet and temperature. We now discuss this point in our revised manuscript.

Point 13: Figure 6H. Why is there a reduction of *Pparg* expression in the *Bin1* kd cells treated with CL?

The reduction in *Pparg* expression is not due to reduced differentiation as lipid accumulation is unaltered (Figure 7G). Furthermore, we did not observe this changes in *Pparg* expression in the *Bin1* knockdown of the D5 clone. Thus, we hypothesize that loss of *BIN1* in *Bin1* expressing cells could have a paracrine effect on neighboring precursors that do not express *Bin1*.

Point 14: Supplementary Figure 6A. How do the authors explain the decrease in *Ucp1* and *Prdm16* expression in the *Tcf25* kd cells vs an increase in *Pgc1a* expression and no difference in mitochondrial uncoupling in the *Tcf25* kd cells (Figure 6C)?

The reduced *Ucp1* and *Prdm16* expression are in line with the observed reduction in the maximal oxygen consumption rate. However, we agree with the reviewer that there are no significant changes in mitochondrial uncoupling, which might require isoproterenol-induced lipolysis to become apparent. Moreover, the upregulation of *Ppargc1a* as well as the white adipocyte marker *Asc1* are puzzling and will require additional functional experiments to characterize the mechanistic role of *TCF25* in brown adipocyte function.

Point 15: The results related to the *Eif5* knockdown cell lines are not clear. According to Figure 6, *shEif5* cells have higher expression of *Ucp1* (Fig. 6H, I, Fig. S6A) and the brown fat marker *P2rx5* (Fig. S6A). However, according to Figure 5 the *Eif5*^{high} population is associated to high *Ucp1* and *P2rx5* expression. The authors should elaborate on this. What would be expected upon *Eif5* overexpression?

Indeed, *Eif5* expression is positively correlated with *Ucp1*, whereas loss of *Eif5* does not reduce thermogenic gene expression or brown fat activity, but it rather enhances it. This is very similar to *CD137*, which, as the reviewer pointed out, is a marker for beige adipocytes, but loss of *CD137* enhances beige adipocyte function (Srivastava et al., PMID: 31919095). We would assume that overexpression of *EIF5* would reduce thermogenic gene expression. However, more mechanistic studies on the function of *EIF5*, such as overexpression in brown adipocytes are required to test this hypothesis.

Point 16: The manuscript would be easier to read if information in the discussion was also included in the respective sections. This applies especially to the known functions of the selected genes and a more detailed explanation of the use of the ProFat database.

We now moved the description of the selected genes and further explanation of the ProFat database into the results part.

Minor comments:

Point 1: More attention should be paid to nomenclature. There are multiple mistakes through the manuscript and the figures concerning the use of capitals, non-capitals and italics when talking about genes (i.e. page 4 '*Ucp1* expression' should be '*Ucp1* expression', page 17 '*P2XR5*' should be '*P2xr5*').

We have corrected these mistakes in the revised version of our manuscript.

Point 2: Figures 1E and 4D are difficult to read; please use lighter colours.

We changed to lighter colors in the revised manuscript.

Point 3: There are several typos and missing words in the manuscript; please correct them.

We apologize for the typographical errors, which we corrected in the revised manuscript.

Referee #2:

Point 1 Editorial request to provide additional qPCR data: In Figures 2 and 3, the authors describe the overall heterogeneity of the 67 cell-lines they obtained, by assessing lipid accumulation and expression of key adipogenic genes such as Pparg, Ucp1, and few additional brown, beige and white markers, revealing quite a large disparity in the differentiation potential of these cells (e.g. Fig. 2A, Fig. 2D, Fig 3A). It would therefore be informative to provide a more comprehensive description of the intrinsic differences of the different clones, to properly interpret the data. This could be addressed by evaluating the expression of additional key genes for adipocyte functions (such as Fabp4, Cd36, Fasn, AdipoQ, Lipe, or others involved in glucose and lipid metabolism, insulin action, and mitochondrial function).

We now show expression of Adiponectin, Cd36, Fabp4, Fasn, Glut4 and Hsl in 64 of the 67 clonal cell lines (Supplementary Figure 2B). We agree with the reviewer that a careful cell biological and biochemical characterization of these 67 clones would provide additional interesting insights into brown adipocyte biology. However, we feel that this is beyond the scope of the current manuscript, where we aim to establish a resource for future studies on murine BAT heterogeneity.

Supplementary Fig. 2B: Quantitative PCR analysis of *Adiponectin*, *Cd36*, *Fabp4*, *Fasn*, *Glut4* and *Hsl* expression in BAT clones. All qPCR data were normalized to *Tbp* and shown as mean \pm SEM (n= 2-3).

Point 2 ^{Editorial request to provide co-stainings:} Figure 5F and 5G describes the cellular distribution of *Eif5*, *Tcf25* and *Bin1* in BAT by immunostaining. However, to investigate how these markers associate with different subcategories of brown adipocytes in vivo (*Ucp1* low to high, Fig. 1 A-B), it would be necessary to perform co-staining with *Ucp1*.

We would like to thank the reviewer for this valuable suggestion. We now include co-immunofluorescence stainings of EIF5, TCF25 and BIN1 with UCP1 (Fig. 6F), which confirm our previous associations of *Eif5*, *Tcf25* and *Bin1* with *Ucp1* expression.

Fig. 6f: BAT co-staining for EIF5, TCF25 or BIN1 (red) and UCP1 (green), with F-actin (gray) and Dapi (blue) from wild-type C57BL/6J mice. Arrows indicate nuclear staining.

Point 3: The authors evaluated the potential functional significance of the 3 selected markers in brown adipocytes following stable knockdown in clonal brown adipocyte cell-lines or in a mixed population. Whereas *Eif5* is associated with higher thermogenic potential (Fig 5), *Eif5* knockdown leads to increased *Ucp1* expression (Supp Fig 6A, Fig 6H), without any change in mitochondrial respiration (Fig 6C and J). On the contrary, *Bin1* is associated with lower thermogenic potential (Fig. 5), and *Bin1* knockdown leads to higher *Ucp1* expression (Supp Fig 6A, Fig 6H), and increased mitochondrial respiration (Fig 6C and J). While these data strongly suggest a direct involvement for the selected markers, especially *Bin1*, in regulating brown adipocyte functions, some interrogations remain. In particular, it is puzzling that the knockdown of *Eif5* or *Bin1*, marking opposite subpopulations, leads to similar effects at the gene expression level, with a different functional outcome. To further evaluate whether the role of *Eif5*, *Tcf25* and *Bin1* in regulating the thermogenic potential of brown adipocytes is cell-intrinsic or rather results from a more complex interplay between different brown adipocyte subpopulations (Fig 6D-J), it would be helpful to test the effect of *Eif5*, *Tcf25* and *Bin1* knockdown in different representative (*Eif5*^{high}, *Tcf25*^{high}, *Bin1*^{high}) clonal cell-lines.

We understand the concern of the reviewer regarding the similar effects resulting from *Eif5* and *Bin1* knockdowns, although they are marking different subpopulations of brown adipocytes. We conclude that EIF5 might have a suppressive effect on thermogenic gene expression itself, similar to what was recently reported for CD137 (Srivastava et al., PMID: 31919095). Thus, it is important to distinguish between the use of a gene or protein as marker for a cell population and the role the protein plays in cellular function.

We performed the knockdown experiment on selected highly *Eif5*, *Tcf25* or *Bin1* expressing clones, as well as in the mixed brown preadipocyte culture, with similar results (Figure 7H and Supplementary Figure 7A). However, to obtain a definitive answer to the reviewers question we would need to investigate the role of these genes *in vivo* either through conditional gene ablation or lineage-tracing. Unfortunately, neither of these genetic tools is available to us and as stated above we repeatedly failed to generate *Eif5*, *Tcf25* and *Bin1* CreERT2 knockin mouse lines. Nevertheless, we feel that our data will provide an important resource and basis for future investigations studying the functional roles of these genes/proteins.

Minor comment:

Point 1: The claim that "loss of *Bin1* results in increased thermogenic gene expression and full mitochondrial uncoupling, at least in clones with high *Bin1* expression" is overstated at this point since *Ucp1* was the only thermogenic gene reported to be regulated, and a unique clonal cell-line was investigated. The authors should provide further data, or need to adjust their conclusion.

We have tuned down our conclusion to “*Ucp1* gene expression”.

Point 2: It would be interesting to know what the authors speculate about the function of these different populations in BAT biology.

We appreciate this question, as usually there is very limited opportunity to speculate. In brief we would speculate that these different kind of brown adipocytes can act to mediate fast or slow responses to environmental cues triggering BAT activation. Furthermore, especially the UCP1 low expressing brown adipocytes could require specific endocrine or environmental inputs for full activation beyond beta-adrenergic stimulation and nutrients. Of even greater importance, however is that if these different brown adipocyte types exist in humans, it would be expected that cellular composition is different between individuals, which could help explain the big variability of humans to having active brown fat, even after cold-exposure.

Referee #3:

Point 1: The authors conclude that the populations identified from the scRNA-seq analysis represent different stages of the differentiation process, rather than distinct lineages. This argument is based solely on computational analyses without any attempts to isolate the populations and study them functionally. The authors should refer to the Merrick et al. Science 2019 paper on how this can be achieved.

Similar to Merrick et al. we performed a computational analysis of the stromal composition based on scRNAseq data. However, unlike Merrick et al. we did not identify cell surface markers of these clusters but rather used clonal cell lines to further dissect the cellular heterogeneity within the brown adipocyte precursor populations. Unfortunately, we identified intracellular marker proteins and therefore cannot use antibody based cell sorting. To overcome this limitation we would need to generate transgenic animals expressing fluorescent proteins under the promoter of these genes. We tried to generate these mice, but were not able to obtain germ line transmission until now. Thus, we hope that our data will serve as a resource for further studies on this subject by us and others.

Point 2 If possible, please include data showing the effects of Eif5High, Tcf25high, and Bin1High cells actually differ functionally, particularly if there is any difference in thermogenesis between these 'sub-populations'- if not please tone down the statement suggesting these to be different lineages or sub-populations in lieu of lacking functional data

Two things about the derived clonal cell lines are not convincing. A) Whether these distinct cell lines reflect cells that are naturally present *in vivo*, and B) whether there are truly functional differences between the cell lines.

Our scRNAseq data and co-immunofluorescence stainings strongly suggest that these cells are present *in vivo*. However, as acknowledged in the manuscript; lineage-tracing experiments using inducible *Eif5*, *Tcf25* and *Bin1* cre-mouse lines would be required to demonstrate the existence of these subpopulations as distinct lineages, as well as to functionally characterize them. As these lines are not available we can only discuss this as a limitation of our study in the manuscript.

Point 3 manuscript text must be edited accordingly: Most importantly, the notion of these cell populations representing distinct lineages is not well supported. Eif5High, Tcf25high, and Bin1High cells *in vivo* may simply reflect cells of distinct "states" rather than developmental lineages. Lineage tracing is needed to support this idea.

We agree with reviewer. We tried to establish cre-lines for the marker genes but were unable to obtain germline transmission. Therefore, *in vivo* lineage tracing experiments were unfortunately not possible. However, we would like to point out that using the clones we are in fact able to trace the development of the cells, which was the bases for identifying EIF5, TCF25 and BIN1. We discuss this limitation of our study in the manuscript.

November 11, 2020

RE: Life Science Alliance Manuscript #LSA-2020-00924-TR

Dr. Siegfried Ussar
Helmholtz Diabetes Center
Institute for Diabetes and Obesity
Ingolstädter Landstraße 1
Neuherberg 85764
Germany

Dear Dr. Ussar,

Thank you for submitting your revised manuscript entitled "Identification and characterization of distinct brown adipocyte subtypes in C57BL/6J mice". We would be happy to publish your paper in Life Science Alliance pending final revisions necessary to meet our formatting guidelines.

Along with the points listed below, please also attend to the following,

- please upload your supplementary figures as single files
- please use the [10 author names, et al.] format in your references (i.e. limit the author names to the first 10)

A. FINAL FILES:

-- Summary blurb (enter in submission system): A short text summarizing in a single sentence the study (max. 200 characters including spaces). This text is used in conjunction with the titles of papers, hence should be informative and complementary to the title. It should describe the context and significance of the findings for a general readership; it should be written in the present tense

and refer to the work in the third person. Author names should not be mentioned.

B. MANUSCRIPT ORGANIZATION AND FORMATTING:

Sincerely,

Shachi Bhatt, Ph.D.
Executive Editor
Life Science Alliance
<https://www.lsjournal.org/>
Tweet @SciBhatt @LSAJournal

November 14, 2020

RE: Life Science Alliance Manuscript #LSA-2020-00924-TRR

Dr. Siegfried Ussar
Helmholtz Diabetes Center
Institute for Diabetes and Obesity
Ingolstädter Landstraße 1
Neuherberg 85764
Germany

Dear Dr. Ussar,

Thank you for submitting your Research Article entitled "Identification and characterization of distinct brown adipocyte subtypes in C57BL/6J mice". It is a pleasure to let you know that your manuscript is now accepted for publication in Life Science Alliance. Congratulations on this interesting work.

DISTRIBUTION OF MATERIALS:

Again, congratulations on a very nice paper. I hope you found the review process to be constructive and are pleased with how the manuscript was handled editorially. We look forward to future exciting submissions from your lab.

Sincerely,

Shachi Bhatt, Ph.D.

Executive Editor

Life Science Alliance

<https://www.lsjournal.org/>
